# FROM LATENT GRAPH TO LATENT TOPOLOGY INFERENCE: DIFFERENTIABLE CELL COMPLEX MODULE

**Claudio Battiloro**[*,1,2]     **Indro Spinelli**[*,1,♯]     **Lev Telyatnikov**[1]
**Michael Bronstein**[3]     **Simone Scardapane**[1,♭,♯]     **Paolo Di Lorenzo**[1,♮,†]

## ABSTRACT

Latent Graph Inference (LGI) relaxed the reliance of Graph Neural Networks (GNNs) on a given graph topology by dynamically learning it. However, most of LGI methods assume to have a (noisy, incomplete, improvable, ...) input graph to rewire and can solely learn regular graph topologies. In the wake of the success of Topological Deep Learning (TDL), we study Latent Topology Inference (LTI) for learning higher-order cell complexes (with sparse and not regular topology) describing multi-way interactions between data points. To this aim, we introduce the Differentiable Cell Complex Module (DCM), a novel learnable function that computes cell probabilities in the complex to improve the downstream task. We show how to integrate DCM with cell complex message-passing networks layers and train it in an end-to-end fashion, thanks to a two-step inference procedure that avoids an exhaustive search across all possible cells in the input, thus maintaining scalability. Our model is tested on several homophilic and heterophilic graph datasets and it is shown to outperform other state-of-the-art techniques, offering significant improvements especially in cases where an input graph is not provided.

## 1 INTRODUCTION

Graph Neural Networks (GNNs) are a versatile tool exploited in a wide range of fields, such as computational chemistry (Gilmer et al., 2017a), physics simulations (Shlomi et al., 2020), and social networks (Xia et al., 2021), just to name a few. GNNs have shown remarkable performance in learning tasks where data are represented over a graph domain, due to their ability to combine the flexibility of neural networks with prior knowledge about data relationships, expressed in terms of the underlying graph topology. The literature on GNNs is extensive and encompasses various techniques, typically categorized into spectral (Bruna et al., 2014) and non-spectral (Gori et al., 2005) methods. The basic idea behind GNNs is to learn node (and/or) edge attributes representations using local aggregation based on the graph topology, i.e. message-passing networks in their most general form (Gilmer et al., 2017b). By leveraging this feature, GNNs have achieved outstanding results in several tasks, including node and graph classification (Kipf & Welling, 2017a), link prediction (Zhang & Chen, 2018), and more specialized tasks such as protein folding (Jumper et al., 2021) and neural algorithmic reasoning (Veličković & Blundell, 2021).

The majority of GNNs assume the graph topology to be fixed (and optimal) for the task at hand, therefore the focus is usually on designing more sophisticated architectures with the aim of improving the message-passing process. Very recently, a series of works (Kazi et al., 2022; de Ocáriz Borde et al., 2023; Topping et al., 2022) started to investigate techniques for Latent Graph Inference (LGI), where the intuition is that data can have some underlying but unknown (latent) graph structure, mainly in cases where only a point cloud of data is available but also when the given graph is suboptimal for the downstream task. LGI is of particular interest on a variety of applications, such as disease prediction (Cosmo et al., 2020; Song et al., 2021), brain network modeling from MRI/fMRI scans (Kan et al., 2022; Qiao et al., 2018), building networks of patients for automatic and personalized diagnosis (Cosmo et al., 2020; Kazi et al., 2019), recommender systems (Zhang et al., 2023), computer vision (Bahl et al., 2022; Raboh et al., 2019), and missing data imputation (Telyatnikov & Scardapane, 2023; Spinelli et al., 2020; You et al., 2020). At the same time, the field of Topological Deep Learning (TDL) (Barbarossa & Sardellitti, 2020; Hajij et al., 2023) started to gain interest, motivated by the fact that many systems are characterized by higher-order interactions that cannot be captured by the

---

*Equal contribution, corresponding authors: {claudio.battiloro, indro.spinelli}@uniroma1.it. Supported by: ♯ PNRR MUR project PE0000013-FAIR; † PNRR MUR project PE000001-RESTART; ♭ Sapienza grant RM1221816BD028D6-DESMOS; ♮ Horizon SNS JU project 6G-GOALS grant Agreement No 101139232.
[1] Sapienza University of Rome, [2] Harvard University, [3] Oxford University.

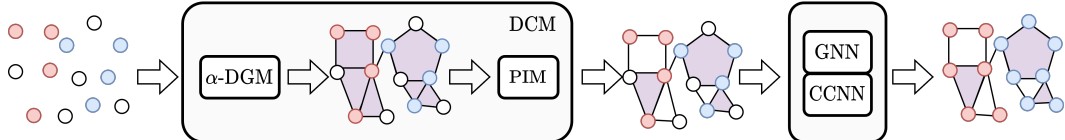

Figure 1: The proposed two-step procedure for Latent Topology Inference (LTI) via regular cell complexes. The Differentiable Cell Complex Module (DCM) is a function that first learns a graph describing the pairwise interactions among data points via the $\alpha$-Differentiable Graph Module ($\alpha$-DGM), and then it leverages the graph as the 1-skeleton of a regular cell complex whose 2-cells (polygons), describing multi-way interactions among data points, are learned via the Polygon Inference Module (PIM). The inferred topology is then used in two message-passing networks, at node (Graph Neural Network, GNN) and edge (Cell Complex Neural Network, CCNN) levels to solve the downstream task. The whole architecture is trained in an end-to-end fashion.

intrinsically pairwise structure of graphs. Topological Neural Networks (TNNs) exploit tools and objects from (algebraic) topology to encode these multi-way relationships, e.g. simplicial (Bodnar et al., 2021b; Giusti et al., 2022; Battiloro et al., 2023e), cell (Bodnar et al., 2021a), or combinatorial complexes (Hajij et al., 2023). However, TDL techniques usually incorporate graphs in higher-order complexes by means of deterministic lifting maps, assigning higher-order cells to cliques or induced cycles of the graph, thus implicitly assuming that these (task-agnostic, deterministic) complexes are optimal for the downstream task. In addition, whenever an input graph is not available, these strategies scale poorly in the size of the input set by requiring an exhaustive (combinatorial) search over all possible cells, making them unfeasible for even medium-sized sets of data points.

**Contribution.** In this paper, we introduce the concept of *Latent Topology Inference* (LTI) by generalizing LGI to higher-order complexes. The goal of LTI is not (only) learning a graph structure describing pairwise interactions but rather learning a higher-order complex describing multi-way interactions among data points. As a first instance of LTI, we introduce the **Differentiable Cell Complex Module (DCM)**, a novel deep learning architecture that dynamically learns a cell complex to improve the downstream task. The DCM implements a two-step inference procedure to alleviate the computational burden: first, learning the 1-skeleton of the complex (i.e., a graph) via a novel improved version of the Differentiable Graph Module (DGM) (Kazi et al., 2022), and then learning which higher-order cells (polygons) should be included in the complex. Both steps leverage message-passing (at node and edge levels) and a sparse sampling technique based on the $\alpha$-entmax class of functions, which allows overcoming the limitation of the original DGM, capable of learning only regular graph topologies. We generalize the training procedure of the DGM (Kazi et al., 2022) to train the DCM in an end-to-end fashion. The DCM is tested on several homophilic and heterophilic datasets and it is shown to outperform other state-of-the-art techniques, offering significant improvements *both when the input graph is provided or not.* In particular, accuracy gains on heterophilic benchmarks with provided input graphs indicate that the DCM leads to robust performance even when the input graph does not fit the data well.

## 1.1 RELATED WORKS

**Latent graph inference** (also referred to as *graph structure learning*) is a problem arising in many applications of graph neural networks and geometric deep learning (Bronstein et al., 2021). Existing LGI approaches belong to two broad classes. **Known input graph.** These approaches assume the input graph is provided but imperfect in some sense, and attempt to modify it to improve the message-passing. Various graph rewiring approaches fall in this category (Topping et al., 2022; Sun et al., 2022; Chen et al., 2020b; Jin et al., 2020; Zhao et al., 2023). **Unknown input graph.** Methods in this class learn the graph structure in a dynamic fashion, without necessarily assuming a fixed graph at the beginning of training. Some approaches assume the latent graph to be *fully connected* (complete), e.g., Transformers (Vaswani et al., 2017) and attentional multi-agent predictive models (Hoshen, 2017), whereas other approaches perform *sparse* latent graph inference, offering computational and learning advantages. Notable sparse LGI techniques are LDS-GNN (Franceschi et al., 2019), Dynamic GCNNs (Wang et al., 2019), and variants of DGM (Kazi et al., 2022; de Ocáriz Borde et al., 2023). A related but different problem is Neural Relational Inference (Kipf et al., 2018), also aiming to infer latent graphs, but usually in an unsupervised fashion, with a focus on physical systems and their time dynamics.

**Topological deep learning** or TDL stems from the pioneering works on Topological Signal Processing (TSP) (Barbarossa & Sardelliitti, 2020; Schaub et al., 2021; Roddenberry et al., 2022; Sardelliitti et al., 2021) that showed the benefits of considering higher-order (multi-way) relationships among data points. Generalizations of the renowned Weisfeiler-Lehman graph isomorphism test to simplicial (SC) (Bodnar et al., 2021b) and cell (CW) (Bodnar et al., 2021a) complexes have been proposed, along with SC and CW message-passing architectures. Convolutional SC and CW architectures have been previously studied in (Ebli et al., 2020; Yang et al., 2022; Hajij et al., 2020; Yang & Isufi, 2023; Roddenberry et al., 2021; Hajij et al., 2022). Attentional SC and CW architectures have been presented in (Battiloro et al., 2023c; Giusti et al., 2022; Goh et al., 2022; Giusti et al., 2023). A notable unifying framework for TDL was proposed in (Hajij et al., 2023), where the concept of combinatorial complex (CC) generalizing SCs, CWs, and hypergraphs, was introduced along with a general class of message-passing CC neural networks. An excellent survey on TDL can be found in (Papillon et al., 2023). Finally, message passing or diffusion on cellular sheaves (Hansen & Ghrist, 2019) built upon graphs were proposed in (Hansen & Gebhart, 2020; Bodnar et al., 2022; Battiloro et al., 2023a;d; Barbero et al., 2022) and shown to be effective in heterophilic settings.

Our paper is related to both classes of works and helps to overcome their limitations via mutual synergy: we significantly improve and generalize DGM to enable, for the first time, latent topology inference and learning of non-regular topologies via the machinery of Topological Deep Learning.

## 2 BACKGROUND

**Regular cell complexes.** We start with the fundamentals of *regular cell complexes*, topological spaces that provide an effective way to represent complex interaction systems of various orders. Regular cell complexes generalize both graphs and simplicial complexes.

***Definition 1 (Regular cell complex)*** (Hansen & Ghrist, 2019; Bodnar et al., 2021a). A *regular cell complex* (CW) is a topological space $\mathcal{C}$ together with a partition $\{\mathcal{X}_\sigma\}_{\sigma \in \mathcal{P}_\mathcal{C}}$ of subspaces $\mathcal{X}_\sigma$ of $\mathcal{C}$ called **cells**, where $\mathcal{P}_\mathcal{C}$ is the indexing set of $\mathcal{C}$, s.t.

1. For each $c \in \mathcal{C}$, every sufficient small neighborhood of $c$ intersects finitely many $\mathcal{X}_\sigma$;
2. For all $\tau, \sigma$ we have that $\mathcal{X}_\tau \cap \overline{\mathcal{X}_\sigma} \neq \varnothing$ iff $\mathcal{X}_\tau \subseteq \overline{\mathcal{X}_\sigma}$, where $\overline{\mathcal{X}_\sigma}$ is the closure of the cell;
3. Every $\mathcal{X}_\sigma$ is homeomorphic to $\mathbb{R}^k$ for some $k$;
4. For every $\sigma \in \mathcal{P}_\mathcal{C}$ there is a homeomorphism $\phi$ of a closed ball in $\mathbb{R}^k$ to $\overline{\mathcal{X}_\sigma}$ such that the restriction of $\phi$ to the interior of the ball is a homeomorphism onto $\mathcal{X}_\sigma$.

From Condition 2, $\mathcal{P}_\mathcal{C}$ has a poset structure, given by $\tau \leq \sigma$ iff $\mathcal{X}_\tau \subseteq \overline{\mathcal{X}_\sigma}$, and we say that $\tau$ *bounds* $\sigma$. This is known as the *face poset* of $\mathcal{C}$. From Condition 4, all of the topological information about $\mathcal{C}$ is encoded in the poset structure of $\mathcal{P}_\mathcal{C}$. Then, a regular cell complex can be identified with its face poset. From now on we will indicate the cell $\mathcal{X}_\sigma$ with its corresponding face poset element $\sigma$. The dimension or order $\dim(\sigma)$ of a cell $\sigma$ is $k$, we call it a $k-$cell and denote it with $\sigma^k$ to make this explicit when necessary. Regular cell complexes can be described via an incidence relation (boundary relation) with a reflexive and transitive closure that is consistent with the partial order introduced in Definition 1. Please see Appendix D for a combinatorial and algebraic description.

***Definition 2 (Boundary Relation).*** We have the boundary relation $\sigma \prec \tau$ iff $\dim(\sigma) \leq \dim(\tau)$ and there is no cell $\delta$ such that $\sigma \leq \delta \leq \tau$.

In other words, Definition 2 states that the boundary of a cell $\sigma^k$ of dimension $k$ is the set of all cells of dimension less than $k$ bounding $\sigma^k$. The dimension or order of a cell complex is the largest dimension of any of its cells. A graph $\mathcal{G}$ is a particular case of a cell complex of order 1, containing only cells of order 0 (nodes) and 1 (edges). We can use the previous definitions to introduce the four types of (local) adjacencies present in regular cell complexes:

***Definition 3 (Cell Complex Adjacencies)*** *(Bodnar et al., 2021a). For a cell complex $\mathcal{C}$ and a cell $\sigma \in \mathcal{P}_\mathcal{C}$, we define:*

- The boundary adjacent cells $\mathcal{B}(\sigma) = \{\tau \mid \tau \prec \sigma\}$, are the lower-dimensional cells on the boundary of $\sigma$. For instance, the boundary cells of an edge are its endpoint nodes.
- The co-boundary adjacent cell $\mathcal{CB}(\sigma) = \{\tau \mid \sigma \prec \tau\}$, are the higher-dimensional cells with $\sigma$ on their boundary. E.g., the co-boundary cells of a node are the edges having that node as an endpoint.

- The lower adjacent cells $\mathcal{N}_d(\sigma) = \{\tau \mid \exists\delta \text{ such that } \delta \prec \sigma \text{ and } \delta \prec \tau\}$, are the cells of the same dimension as $\sigma$ that share a lower dimensional cell on their boundary. The line graph adjacencies between the edges are a typical example of this.

- The upper adjacent cells $\mathcal{N}_u(\sigma) = \{\tau \mid \exists\delta \text{ such that } \sigma \prec \delta \text{ and } \tau \prec \delta\}$. These are the cells of the same dimension as $\sigma$ that are on the boundary of the same higher-dimensional cell.

***Definition 4 (k-skeleton)***. A *k-skeleton* of a regular cell complex $\mathcal{C}$ is the subcomplex of $\mathcal{C}$ consisting of cells of dimension at most $k$.

From Definition 1 and Definition 5, it is clear that the 0-skeleton of a cell complex is a set of vertices and the 1-skeleton is a set of vertices and edges, thus a graph. For this reason, given a graph $\mathcal{G} = \{\mathcal{V}, \mathcal{E}\}$, it is possible to build a regular cell complex "on top of it", i.e. a cell complex $\mathcal{C}_\mathcal{G}$ whose 1-skeleton is isomorphic to $\mathcal{G}$.

**Remark 1.** We remark that the term "regular" is ambiguous in this context, because a regular cell complex is an object as in Definition 1, while a regular graph is a graph whose neighborhoods' cardinalities are all the same. For this reason, in the following, we will refer to regular cell complex(es) simply as "cell complex(es)," reserving the term "regular" only for graphs.

In this work, we attach order 2 cells to an inferred (learned) subset of induced cycles of the graph $\mathcal{G}$ up to length $K_{max}$, we refer to them as *polygons*, and we denote the resulting order-2 cell complex with $\mathcal{C}_\mathcal{G} = \{\mathcal{V}, \mathcal{E}, \mathcal{P}\}$, where $\mathcal{P}$ is the polygons set, with $|\mathcal{V}| = N$, $|\mathcal{E}| = E$, and $|\mathcal{P}| = P$. This procedure is formally a skeleton-preserving lifting map; a detailed discussion about the lifting of a graph into a cell complex can be found in (Bodnar et al., 2021a).

**Signals over Cell Complexes** A $k$-cell signal is defined as a mapping from the set $\mathcal{D}_k$ of all $k$-cells contained in the complex, with $|\mathcal{D}_k| = N_k$, to real numbers (Sardellitti et al., 2021):

$$x_k : \mathcal{D}_k \to \mathbb{R} \tag{1}$$

Therefore, for an order-2 complex $\mathcal{C}_\mathcal{G}$, the $k$-cell signals are defined as the following mappings:

$$x_0 : \mathcal{V} \to \mathbb{R}, \qquad x_1 : \mathcal{E} \to \mathbb{R}, \qquad x_2 : \mathcal{P} \to \mathbb{R}, \tag{2}$$

representing node, edge, and polygon signals, respectively, with $N_0 = N$, $N_1 = E$, and $N_2 = P$. If $F$ $k-$cell signals are available, we collect them in a feature matrix $\mathbf{X}_{k,in} = \{\mathbf{x}_{k,in}(i)\}_{i=1}^{N_k} \in \mathbb{R}^{N_k \times F}$, where the $i$-th row $\mathbf{x}_{k,in}(i) = [x_{k,1}(\sigma_i^k), \dots, x_{k,F}(\sigma_i^k)] \in \mathbb{R}^F$ collects the features, i.e. the signals values, of the $i$-th $k-$cell $\sigma_i^k$. The definition of $k-$cell signal in (1) is sufficient and rigorous for the scope of this paper, however more formal topological characterizations in terms of chain and cochain spaces can be given (Sardellitti et al., 2021; Bodnar et al., 2021a; Hajij et al., 2023).

## 3 A Differentiable Layer for Latent Topology Inference

We propose a novel layer, depicted at high level in Figure 1 and in detail in Figure 2, comprising of a series of modules among which the most important one is the *Differentiable Cell Complex Module* (DCM), the first fully differentiable module that performs LTI, i.e. learns a cell complex describing the hidden higher-order interactions among input data points. The DCM is equipped with a novel sampling algorithm that, at the best of our knowledge, is the first graph/complex sampling technique that allows to generate topologies whose neighborhoods cardinality is not fixed a priori but can be freely learned in an end-to-end fashion.

The proposed layer takes as input the node feature matrix $\mathbf{X}_{0,in} \in \mathbb{R}^{N \times F_{in}}$, and gives as output the updated node feature matrix $\mathbf{X}_{0,out} \in \mathbb{R}^{N \times F_{out}}$ and the inferred latent cell complex $\mathcal{C}_{\mathcal{G}_{out}}$. Optionally, the layer can take as input also a graph $\mathcal{G}_{in}$. To make the layer self-consistent in a multi-layer setting, the output can be reduced to the updated node feature matrix $\mathbf{X}_{0,out}$ and the 1-skeleton (graph) $\mathcal{G}_{out}$ of $\mathcal{C}_{\mathcal{G}_{out}}$.

We employ a two-step inference procedure to keep the computational complexity tractable, i.e. the DCM module first samples the 1-skeleton of the latent cell complex (possibly in a sparse way, i.e. $E << N^2$), and then it samples the polygons among the induced cycles generated by the sampled edges. The first step is implemented via the novel $\alpha$-Differentiable Graph Module ($\alpha-$DGM), while the second step is implemented via the novel Polygon Inference Module (PIM). Directly sampling among all the possible polygons, thus trivially generalizing the DGM framework (Kazi et al., 2022),

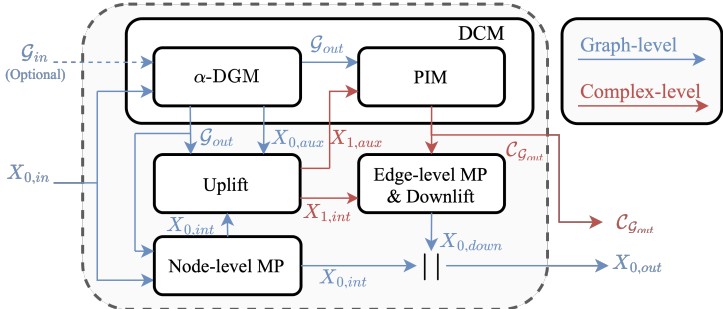

Figure 2: The DCM and the proposed architecture.

would have led to intractable complexity, e.g. even at triangles level the sampling would have had a complexity of the order of $\mathcal{O}(\sqrt{E^3}) = \mathcal{O}(N^3)$, being all edges candidate to be sampled. In the following, we describe in detail each module of the proposed layer.

### 3.1 THE $\alpha$-DIFFERENTIABLE GRAPH MODULE

The $\alpha$-DGM is a novel variation of the DGM and it is responsible for inferring the 1-skeleton (graph) of the latent cell complex. One of the main limitations of the DGM (Kazi et al., 2022) is the constraint of inferring only regular graphs; we solve this problem by proposing a novel sampling procedure based on the $\alpha-$entmax class of functions (Peters et al., 2019), recently introduced in the NLP community to obtain sparse alignments and zero output probabilities. With fixed $\alpha > 1$ and input $\mathbf{s} \in \mathbb{R}^d$, the $\alpha-$entmax function $\boldsymbol{\alpha}_{ENT} : \mathbb{R}^d \to \Delta^d$ is defined as:

$$\boldsymbol{\alpha}_{ENT}(\mathbf{s}) = \arg\max_{\mathbf{u} \in \Delta^d} \langle \mathbf{u}, \mathbf{s} \rangle_{\mathbb{R}^d} + H_\alpha(\mathbf{u}), \tag{3}$$

where $\Delta^d = \{\mathbf{u} \in \mathbb{R}^d : \mathbf{u} \geq 0, \|\mathbf{u}\|_1 = 1\}$ is the $d$-dimensional probability simplex and $H_\alpha$ is the family of Tsallis $\alpha$-entropies parametrized by a parameter $\alpha \geq 1$ (Peters et al., 2019). For $\alpha = 1$, equation (3) recovers the standard softmax, while for $\alpha > 1$ solutions can be sparse with a degree depending on $\alpha$. In practice, $\alpha$ can be initialized at a reasonable value (e.g., 1.5) and adapted via gradient descent (Correia et al., 2019).

In $\alpha$-DGM, we first compute auxiliary node features $\mathbf{X}_{0,aux} = \nu_0(\mathbf{X}_{0,in}) \in \mathbb{R}^{N \times d_0}$, where $\nu_0(\cdot)$ is a learnable function, e.g. a GNN if an initial graph $\mathcal{G}_{in}$ is provided or an MLP otherwise. At this point, a (pseudo-)similarity function $\text{sim}(\cdot)$ is chosen, the similarities among node embeddings are computed and collected in the vectors $\mathbf{z}_i = \{\text{sim}(\mathbf{x}_{0,aux}(i), \mathbf{x}_{0,aux}(j))\}_j \in \mathbb{R}^N$, $i, j = 1, \ldots, N$. Valid examples of $\text{sim}(\cdot)$ are cosine similarity, inverse or minus Euclidean square distance.

The (sparse) edge probabilities are then obtained node-wise via $\alpha-$entmax, i.e. the vectors $\mathbf{p}_i = \boldsymbol{\alpha}_{ENT}(\mathcal{LN}(\mathbf{z}_i)) \in \mathbb{R}^N$, $i = 1, \ldots, N$ are computed; layer normalization $\mathcal{LN}$ is employed to have better control on the similarities statistics and, consequently, more stability in the training procedure. The $(i, j)$ edge is included in the inferred graph if $\mathbf{p}_i(j) > 0$. This procedure leads to a directed graph that would be incompatible with the cell complex structure introduced in Section 2, whose 1-skeleton is an undirected graph; the inferred directed graph is converted to the closest undirected graph $\mathcal{G}_{out} = \{\mathcal{N}, \mathcal{E}\}$, i.e. each inferred edge is considered as a bidirectional edge to obtain $\mathcal{E}$. We want further stress the fact that computing edge probabilities via the $\alpha$-entmax leads to sparse and non-regular graphs whose sparsity level can be controlled (or learned) tuning the parameter $\alpha$. A detailed pictorial description of the $\alpha$-DGM is shown in Figure 3 (left).

**Remark 2.** The choice of applying the $\alpha$-entmax in a node-wise fashion and not directly on a vector containing all the possible similarities (that would have naturally led to an undirected graph) is due to the fact that, in the latter case, we empirically observed that the $\alpha$-entmax consistently leads to heavily disconnected graphs with few hub nodes (therefore, also leading to performance drops).

At this point, the intermediate node features $\mathbf{X}_{0,int} \in \mathbb{R}^{N \times F_{int}}$ are computed with one or more usual message-passing (MP) rounds over the inferred graph:

$$\mathbf{x}_{0,int}(i) = \gamma_0\left(\mathbf{x}_{0,in}(i), \bigoplus_{j \in \mathcal{N}_u(\sigma_i^0)} \phi_0^{\mathcal{N}_u}\left(\mathbf{x}_{0,in}(i), \mathbf{x}_{0,in}(j)\right)\right), \tag{4}$$

where $\gamma_0(\cdot)$ and $\phi_0^{\mathcal{N}_u}(\cdot)$ are learnable functions and $\bigoplus$ is any aggregation operator.

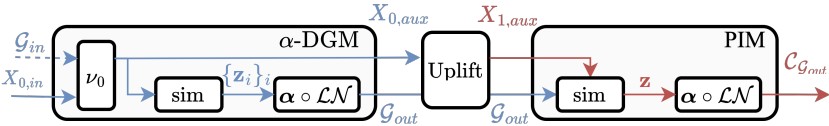

Figure 3: The $\alpha$-Differentiable Graph Module (left) and the Polygon Inference Module (right).

## 3.2 UPLIFT MODULE

To fully exploit the potential of cell complexes, it is not sufficient to work at the node level. For this reason, intermediate edge features are learned (or computed) with a MP round of the form:

$$\mathbf{x}_{1,int}(i) = \psi_1\left(\bigoplus_{j\in\mathcal{B}(\sigma_i^1)} \phi_1^{\mathcal{B}}\Big(\mathbf{x}_{0,int}(j)\Big)\right), \tag{5}$$

where $\psi_1(\cdot)$ is a (possibly) learnable function. The boundary $\mathcal{B}(\sigma_i^1)$ of edge $i$ is composed by its endpoints nodes $\sigma_j^0$ and $\sigma_v^0$. Therefore, the intermediate features $\mathbf{x}_{1,int}(i)$ of edge $i$ are learned (or computed) as a function of the intermediate node features $\mathbf{x}_{0,int}(j)$ and $\mathbf{x}_{0,int}(v)$.

## 3.3 THE POLYGON INFERENCE MODULE

The Polygon Inference Module (PIM) is a novel module responsible of inferring the polygons of the latent cell complex by sampling a subset of the induced cycles of the inferred 1-skeleton $\mathcal{G}_{out}$.

To this aim, auxiliary edge features are computed from the auxiliary node features with a MP round as in (5), which we denote in this context with $\mathbf{X}_{1,aux} = \nu_1(\mathbf{X}_{0,aux}) \in \mathbb{R}^{E\times d_1}$ for notation consistency.

At this point, we need a similarity function for the edge embeddings belonging to the same induced cycle. We decide to use the sum of the pairwise similarities, e.g. for a generic cycle of length $k$ made by the edges whose indices are collected in a index set $\mathcal{I}$, the similarity is computed (with a slight abuse of notation) as:

$$\text{sim}(\{\mathbf{x}_{1,aux}(i)\}_{i\in\mathcal{I}}) = \sum_{i\in\mathcal{I}}\sum_{j\in\mathcal{I},j\neq i} \text{sim}(\mathbf{x}_{1,aux}(i), \mathbf{x}_{1,aux}(j)). \tag{6}$$

The similarity in (6) for an induced cycle of length $k \leq K_{max}$ contains $\frac{k!}{(k-2)!2!}$ terms, however, $K_{max}$ is usually a very small integer and the sum can be trivially distributed. In general, $\text{sim}(\{\mathbf{x}_{1,aux}(i)\}_{i\in\mathcal{I}})$ doesn't need to be the sum of pairwise similarities. It can be arbitrarily designed, as long as it is a similarity measure, i.e. it is higher if the involved embeddings are similar.

The similarities are collected in a vector $\mathbf{z} \in \mathbb{R}^{\widetilde{P}}$, where $\widetilde{P}$ is the number of induced cycles, and the polygons probabilities are computed as $\mathbf{p} = \boldsymbol{\alpha}_{ENT}(\mathcal{LN}(\mathbf{z})) \in \mathbb{R}^{\widetilde{P}}$. We set $\mathcal{C}_{\mathcal{G}_{out}} = \{\mathcal{N}, \mathcal{E}, \mathcal{P}\}$, where $\mathcal{P}$ are the induced cycles with positive probabilities and $|\mathcal{P}| = P \leq \widetilde{P}$ (possibly $P << \widetilde{P}$).

The updated edge features $\mathbf{X}_{1,out} \in \mathbb{R}^{E\times F_{out}}$ are computed with one or more MP rounds as:

$$\mathbf{x}_{1,out}(i) = \gamma_1\left(\mathbf{x}_{1,int}(i), \bigotimes_{\mathcal{N}_k\in\{\mathcal{N}_d,\mathcal{N}_u\}} \bigoplus_{j\in\mathcal{N}_k(\sigma_i^1)} \phi_1^{\mathcal{N}_k}\Big(\mathbf{x}_{1,int}(i), \mathbf{x}_{1,int}(j)\Big)\right), \tag{7}$$

where $\bigotimes$ is a neighborhoods aggregation operator (Hajij et al., 2023), $\gamma_1(\cdot)$, $\phi_1^{\mathcal{N}_d}(\cdot)$, and $\phi_1^{\mathcal{N}_u}(\cdot)$ are learnable functions. A detailed pictorial description of the PIM is shown in Figure 3 (right).

**Remark 3.** The message-passing rounds in (7) and in (5) (as a special case) are instances of message-passing neural networks over cell complexes. Several MP schemes can be defined for cell (simplicial, combinatorial) complexes, please refer to (Bodnar et al., 2021a; Hajij et al., 2023) for more details. We employ the schemes in (5)-(7) for two reasons: the first one is that using more sophisticated MP rounds (e.g. moving to cells of higher order than polygons or designing more intensive messages exchange among different cell orders) would have lead to computational intractability; the second one is that moving to higher order cells also introduces a series of tricky theoretical issues (in terms of the structure of the complex) that are not trivial to tackle (Hansen & Ghrist, 2019). We plan to investigate these directions in future works.

## 3.4 DOWNLIFT MODULE AND OUTPUT COMPUTATION

At this point, the output node features $\mathbf{X}_{0,out} \in \mathbb{R}^{N \times F_{out}}$ are learned (or computed) from the updated edge features $\mathbf{X}_{1,out} \in \mathbb{R}^{E \times F_{out}}$ and the intermediate node features $\mathbf{X}_{0,int} \in \mathbb{R}^{N \times F_{int}}$ as:

$$\mathbf{x}_{0,out}(i) = \left[\mathbf{x}_{0,int}(i) \,\middle\|\, \mathbf{x}_{0,down}(i)\right], \tag{8}$$

where the $\mathbf{X}_{0,down} \in \mathbb{R}^{N \times F_{out}}$ are obtained with a MP round of the form:

$$\mathbf{x}_{0,down}(i) = \psi_0\left(\bigoplus_{j \in \mathcal{CB}(\sigma_i^0)} \phi_0^{\mathcal{CB}}\left(\mathbf{x}_{1,out}(j)\right)\right) \tag{9}$$

with $\psi_0(\cdot)$ being a (possibly) learnable function. The coboundary $\mathcal{CB}(\sigma_i^0)$ of node $i$ is composed by all the edges for which node $i$ is an endpoint. Therefore, the output features $\mathbf{x}_{0,out}(i)$ of node $i$ are learned (or computed) as the concatenation of its intermediate features and features obtained downlifting the updated features of the edges for which $i$ is an endpoint.

We present a comparison in terms of computational complexity between the DCM and the DGM (Kazi et al., 2022) in Appendix B .

## 3.5 TRAINING OF THE DIFFERENTIABLE CELL COMPLEX MODULE

The proposed sampling scheme based on the $\alpha$-entmax does not allow the gradient of the downstream task loss to flow both through the graph and polygons inference branches of the DCM, due to the fact that they involve only auxiliary features and the entmax outputs are substantially just a way of indexing the edges and the polygons present in the inferred complex. To enable a task-oriented end-to-end training of the DCM, we generalize the approach of DGM (Kazi et al., 2022; de Ocáriz Borde et al., 2023) and design an additional loss term that rewards edges and polygons involved in a correct classification (in classification tasks) or in "good" predictions (in regression tasks), and penalizes edges and polygons that lead to misclassification or high error predictions. Refer to Appendix A for the details.

## 4 EXPERIMENTAL RESULTS

In this Section, we evaluate the effectiveness of the proposed framework on several heterophilic and homophilic graph benchmarks. Graph homophily is the principle that nodes with similar labels are more likely to be connected. Traditional Graph/Cell Complex Convolutional Neural Networks (GCNs and CCCNs) implicitly rely on the homophily assumption, and performance drops are typically observed in heterophilic settings (Bodnar et al., 2022; Spinelli et al., 2022; 2023).

Our main goal is to show that *latent topology inference* via the differentiable cell complex module (DCM) allows learning higher-order relationships among data points that lead to significant improvements w.r.t. *latent graph inference*. Since DCM is a generalization of the differentiable graph module (DGM), we use as a comparison the original (discrete) DGM (Kazi et al., 2022) (denoted DGM-E) and its recently introduced non-Euclidean version (de Ocáriz Borde et al., 2023) (denoted DGM-M). Moreover, we also report the results of a simplified variant of our model (denoted as DCM ($\alpha = 1$)), in which the graph is explicitly learned and all the polygons are taken, i.e. $\alpha = 1$ in the PIM; in this (lower complexity) case, the $\alpha$-DGM guides both the edges (explicitly) and the polygons (implicitly) inference steps. We also test a variant, denoted with GCN-CCCN, where the complex is not imputed at all, i.e. the input graph is used in a GCN and all the polygons are used in a CCCN, and a variant, denoted with KCM, where the graph is imputed using a KNN after embedding the node features using an MLP/GCN, and all the polygons are used. Finally, we report the vanilla MLP and GCN (Kipf & Welling, 2017b), GCN2 (Chen et al., 2020a), GAT (Veličković et al., 2018), and (convolutional) CWN (Bodnar et al., 2021a) as further baselines and comparisons. In all the experiments, we utilize GCNs at the graph level and CCCNs at the edge level; the details about the architectures employed to obtain the results are given in Appendix F. The **first** and the **second** best results are highlighted.

We follow the same core experimental setup of (de Ocáriz Borde et al., 2023) on transductive classification tasks; in particular, we first focus on standard graph datasets such as *Cora*, *CiteSeer*

Table 1: Homophilic-graph node classification benchmarks. Test accuracy in % avg.ed over 10 splits.

| | | Cora | CiteSeer | PubMed | Physics | CS |
|---|---|---|---|---|---|---|
| | Model/Hom. level | 0.81 | 0.74 | 0.80 | 0.93 | 0.80 |
| w/o graph | MLP | $58.92 \pm 3.28$ | $59.48 \pm 2.14$ | $85.75 \pm 1.02$ | $94.91 \pm 0.28$ | $87.80 \pm 1.54$ |
| | KCM | $78.47 \pm 2.09$ | $75.20 \pm 2.41$ | $86.66 \pm 0.91$ | $95.61 \pm 0.18$ | $95.14 \pm 0.32$ |
| | DGM-E | $62.48 \pm 3.24$ | $62.47 \pm 3.20$ | $83.89 \pm 0.70$ | $94.03 \pm 0.45$ | $76.05 \pm 6.89$ |
| | DGM-M | $70.85 \pm 4.30$ | $68.86 \pm 2.97$ | $\mathbf{87.43} \pm 0.40$ | $95.25 \pm 0.36$ | $92.22 \pm 1.09$ |
| | DCM | $\mathbf{78.80} \pm 1.84$ | $\mathbf{76.47} \pm 2.45$ | $87.38 \pm 0.91$ | $\mathbf{96.45} \pm 0.12$ | $\mathbf{95.40} \pm 0.40$ |
| | DCM ($\alpha = 1$) | $\mathbf{78.73} \pm 1.99$ | $\mathbf{76.32} \pm 2.75$ | $\mathbf{87.47} \pm 0.77$ | $\mathbf{96.22} \pm 0.27$ | $\mathbf{95.35} \pm 0.37$ |
| w graph | GCN | $83.11 \pm 2.29$ | $69.97 \pm 2.00$ | $85.75 \pm 1.01$ | $95.51 \pm 0.34$ | $87.28 \pm 1.54$ |
| | GCN2 | $\mathbf{87.85} \pm 1.41$ | $78.53 \pm 2.66$ | $\mathbf{89.60} \pm 0.70$ | $\mathbf{97.41} \pm 0.34$ | $95.05 \pm 0.38$ |
| | GAT | $\mathbf{89.81} \pm 1.77$ | $78.18 \pm 2.31$ | $88.53 \pm 0.61$ | $\mathbf{98.87} \pm 0.30$ | $94.42 \pm 0.70$ |
| | KCM | $78.43 \pm 2.11$ | $75.23 \pm 2.45$ | $86.61 \pm 0.95$ | $96.16 \pm 0.17$ | $95.46 \pm 0.36$ |
| | CWN | $88.63 \pm 1.91$ | $75.53 \pm 2.13$ | $87.97 \pm 0.77$ | $96.23 \pm 0.24$ | $93.52 \pm 0.59$ |
| | GCN+CCCN | $86.09 \pm 1.82$ | $78.36 \pm 3.33$ | $88.59 \pm 0.67$ | $96.90 \pm 0.30$ | $95.31 \pm 0.49$ |
| | DGM-E | $82.11 \pm 4.24$ | $72.35 \pm 1.92$ | $87.69 \pm 0.67$ | $95.96 \pm 0.40$ | $87.17 \pm 3.82$ |
| | DGM-M | $86.63 \pm 3.25$ | $75.42 \pm 2.39$ | $87.82 \pm 0.59$ | $96.21 \pm 0.44$ | $92.86 \pm 0.96$ |
| | DCM | $85.78 \pm 1.71$ | $\mathbf{78.72} \pm 2.84$ | $88.49 \pm 0.62$ | $96.99 \pm 0.44$ | $\mathbf{95.79} \pm 0.48$ |
| | DCM ($\alpha = 1$) | $85.97 \pm 1.86$ | $\mathbf{78.60} \pm 3.16$ | $\mathbf{88.61} \pm 0.69$ | $96.69 \pm 0.46$ | $\mathbf{95.78} \pm 0.49$ |

Table 2: Heterophilic-graph node classification benchmarks. Test accuracy in % avg.ed over 10 splits.

| | | Texas | Wisconsin | Squirrel | Chameleon |
|---|---|---|---|---|---|
| | Model/Hom. level | 0.11 | 0.21 | 0.22 | 0.23 |
| w/o graph | MLP | $77.78 \pm 10.24$ | $85.33 \pm 4.99$ | $30.44 \pm 2.55$ | $40.35 \pm 3.37$ |
| | KCM | $84.12 \pm 11.37$ | $87.10 \pm 5.15$ | $35.15 \pm 1.38$ | $52.12 \pm 2.02$ |
| | DGM-E | $80.00 \pm 8.31$ | $\mathbf{88.00} \pm 5.65$ | $34.35 \pm 2.34$ | $48.90 \pm 3.61$ |
| | DGM-M | $81.67 \pm 7.05$ | $\mathbf{89.33} \pm 1.89$ | $35.00 \pm 2.35$ | $48.90 \pm 3.61$ |
| | DCM | $\mathbf{85.71} \pm 7.87$ | $87.49 \pm 5.94$ | $\mathbf{35.55} \pm 2.24$ | $\mathbf{53.63} \pm 3.07$ |
| | DCM ($\alpha = 1$) | $\mathbf{84.96} \pm 10.24$ | $86.72 \pm 6.02$ | $\mathbf{35.25} \pm 2.22$ | $\mathbf{53.67} \pm 3.19$ |
| w graph | GCN | $41.66 \pm 11.72$ | $47.20 \pm 9.76$ | $24.19 \pm 2.56$ | $32.56 \pm 3.53$ |
| | GCN2 | $75.50 \pm 7.81$ | $74.57 \pm 5.38$ | $33.09 \pm 1.76$ | $49.50 \pm 3.02$ |
| | GAT | $66.72 \pm 11.22$ | $60.52 \pm 9.23$ | $\mathbf{35.07} \pm 2.13$ | $50.73 \pm 3.12$ |
| | KCM | $83.92 \pm 11.15$ | $84.92 \pm 5.21$ | $34.47 \pm 1.49$ | $\mathbf{53.12} \pm 2.02$ |
| | CWN | $65.87 \pm 6.33$ | $64.57 \pm 7.12$ | $32.44 \pm 2.75$ | $43.86 \pm 2.51$ |
| | GCN+CCCN | $84.43 \pm 9.11$ | $84.03 \pm 5.42$ | OOM | OOM |
| | DGM-E | $60.56 \pm 8.03$ | $70.67 \pm 10.49$ | $29.87 \pm 2.46$ | $44.19 \pm 3.85$ |
| | DGM-M | $62.78 \pm 9.31$ | $76.00 \pm 3.26$ | $30.44 \pm 2.38$ | $45.68 \pm 2.66$ |
| | DCM | $\mathbf{84.87} \pm 10.04$ | $\mathbf{86.33} \pm 5.14$ | $34.95 \pm 2.59$ | $53.05 \pm 3.00$ |
| | DCM ($\alpha = 1$) | $\mathbf{84.96} \pm 5.60$ | $\mathbf{85.36} \pm 5.05$ | $\mathbf{35.13} \pm 2.27$ | $\mathbf{53.76} \pm 3.72$ |

(Yang et al., 2016), *PubMed*, *Physics* and *CS* (Shchur et al., 2019), which have high homophily levels ranging from 0.74 to 0.93. We then test our method on several challenging heterophilic datasets, *Texas*, *Wisconsin*, *Squirrel*, and *Chameleon* (Rozemberczki et al., 2021), which have low homophily levels ranging between 0.11 and 0.23. The results for the homophilic and heterophilic datasets are presented in Tables 1 and 2, respectively. All the models were tested in two settings: assuming the original graph is available (marked *w graph* in Tables 1 and 2) and a more challenging case in which the input graph is assumed to be unavailable (*w/o graph*).

From Tables 1 and 2, it is evident that the DCM exhibits consistently superior performance compared to alternative methods across both homophilic and heterophilic datasets, both with and without provided input graphs. As expected, our method achieves top performance without an input graph for the heterophilic datasets and with an input graph for the homophilic datasets. We achieve remarkable results considering the input graph for heterophilic datasets. Despite exposing the model to the "wrong" input graphs (in the sense that the topology does not match the structure of the node features), the overall performance of the DCM remains stable, yielding remarkable improvements exceeding 20% for Texas and approximately 10% for Wisconsin compared to DGM-M (de Ocáriz Borde et al., 2023). Therefore, observing the results across both homophilic and heterophilic datasets when the input graph is provided, we can appreciate how DCM exploits the available "good" graphs in the first case while being less sensitive to the "wrong" ones in the latter. Our performance on the heterophilic datasets suggests that the learned latent complex has a homophilic 1-skeleton (graph) that enables the involved GCNs and CCCNs to reach higher accuracies. To corroborate this hypothesis, in Figure 4, we show the evolution of the latent topology during the training for the Texas dataset along with the nodes degree distribution, the percentage of sampled polygons $\%_p$, and the homophily level $h$ that we note reaching 0.99 on the final inferred graph from the initial 0.11. Moreover, the fact that most of the inferred polygons belong to the same class and the pretty high spread of the degree distribution further confirm the effectiveness of the proposed architecture. In Appendix,E and C, we report ablation studies, plots, and interpretability hints on the inferred complexes.

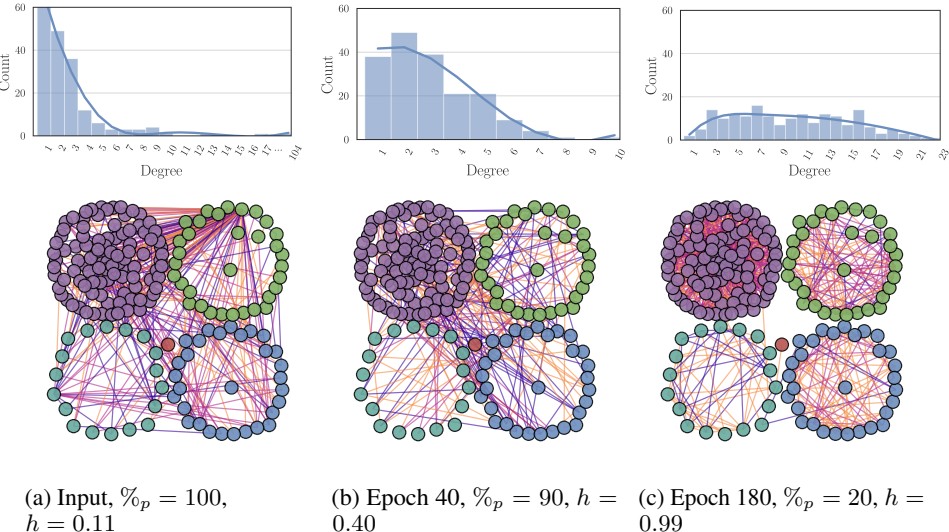

(a) Input, %_p = 100, h = 0.11     (b) Epoch 40, %_p = 90, h = 0.40    (c) Epoch 180, %_p = 20, h = 0.99

Figure 4: Evolution of the latent complex for the Texas dataset, along with homophily level and nodes degree distribution. Edges in orange, triangles in lilac, squares in purple ($K_{max} = 4$)

## 5    CONCLUSIONS

We introduced the paradigm of latent topology inference, aiming to not (only) learn a latent graph structure but rather higher-order topological structures describing multi-way interactions among data points. We made LTI implementable by introducing the Differentiable Cell Complex Module, a novel learnable function that dynamically learns a regular cell complex to improve the downstream task in an end-to-end fashion and in a computationally tractable way, via to a two-step inference procedure that avoids an exhaustive search across all possible cells in the input. We showed the effectiveness of the DCM on a series of homophilic and heterophilic graph benchmarks datasets, comparing its performance against state-of-the-art latent graph inference methods, and showing its competitive performance both when the input graph is provided or not.

## 6    LIMITATIONS AND FUTURE DIRECTIONS

To our knowledge, DCM is the first differentiable approach for latent topology inference. The promising results open several avenues for future work.

**Methodological.** Although cell complexes are very flexible topological objects, other instances of LTI could leverage hypergraphs or combinatorial complexes (CCs) (Hajij et al., 2022), which are able to handle both hierarchical and set-type higher-order interactions. Second, remaining within cell complexes, different MP schemes (Hajij et al., 2023), lifting maps (Bodnar et al., 2021a), or model spaces (de Ocáriz Borde et al., 2023) are of interest for future work. Third, extending our framework to weighted (Battiloro et al., 2023b) or directed complexes (Courtney & Bianconi, 2018) could give further insights. Finally, one potentially interesting direction is merging LTI and Sheaf Neural Networks in order to learn cellular sheaves defined on latent higher-order complexes.

**Computational.** While most of our experimental validation of the DCM focused on transductive tasks, we could also tackle inductive ones (de Ocáriz Borde et al., 2023). Future works could tailor the DCM to fit challenging specific applications as the ones presented in the introduction for LGI. Moreover, though computationally tractable thanks to the proposed two-step inference procedure, DCM may benefit from further improvements to effectively scale on very large datasets. The main bottlenecks are MP operations and the search for the induced cycles of the learned graph. A possible solution for the former could be neighbor samplers (de Ocáriz Borde et al., 2023), while the latter could be tackled by leveraging stochastic search methods or moving to more flexible topological spaces, e.g. CCs. Finally, our sampling strategy implicitly assumes that there are no specific edges and polygons distribution requirements, e.g. a sampling budget is given or a particular correlation structure needs to be imposed on the cells. In these cases, incorporating more sophisticated sampling methods like IMLE (Li & Malik, 2018; Serra & Niepert, 2022) could be beneficial.

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

## REPRODUCIBILITY STATEMENT

We include all the details about our experimental setting, including the choice of hyperparameters and the specifications of our machine, in Appendix F. We provide all the code, data splits, and virtual environment needed to replicate the experiments at the following anonymized repository: `https://github.com/spindro/differentiable_cell-complex_module`.

## A  TRAINING PROCEDURE

Here we discuss the key concepts about the training of the DCM. We detail how to back-propagate through the entmax-based discrete sampling at the 1-cell (edge) level (in the $\alpha-$DGM) and at the 2-cell (polygon) level (in the PIM). We follow the approach proposed by(Kazi et al., 2022) and introduce a supplementary loss that rewards edges and polygons involved in a correct classification and penalizes the ones which result in misclassification. We define the reward function:

$$\delta\left(y_i, \hat{y}_i\right) = \mathbb{E}\left(a_i\right) - a_i\,, \tag{10}$$

as the difference between the average accuracy of the $i$-th sample and the current prediction accuracy, where $y_i$ and $\hat{y}_i$ are the predicted and true labels, and $a_i = 1$ if $y_i = \hat{y}_i$ or 0 otherwise. We then define the loss associated to the edge sampling as follows:

$$L_{GL} = \sum_{i=1}^{N}\left(\delta\left(y_i, \hat{y}_i\right)\sum_{j=1}^{N}\mathbf{p}_i(j)\right) \tag{11}$$

We estimate $\mathbb{E}\left(a_i\right)$ with an exponential moving average on the accuracy during the training process

$$\mathbb{E}\left(a_i\right)^{(t+1)} = \mu\mathbb{E}\left(a_i\right)^{(t)} + (1-\mu)a_i\,, \tag{12}$$

with $\mu = 0.9$ in all our experiments and $\mathbb{E}\left(a_i\right)^{(0)} = \frac{1}{\#\,\text{classes}}$.

Similarly, the loss associated with the polygon sampling is defined as follows:

$$L_{PL} = \sum_{i=1}^{N}\left(\delta\left(y_i, \hat{y}_i\right)\sum_{j\in\mathcal{P}_i}\mathbf{p}(j)\right)\,, \tag{13}$$

where $\mathcal{P}_i$ is the set of indices of the polygons for which node $i$ is a vertex. Following (de Ocáriz Borde et al., 2023), the accuracy can be replaced with the R2 score in the case of regression tasks. In every experiment, we set the initial value of $\alpha$ to 1.5.

Table 3: Execution times expressed in Seconds.

| | | Graph MP | Graph Sampling | Lifting | Cell Sampling | Cell MP |
|---|---|---|---|---|---|---|
| **Cora** | GCN | $5e^{-5}$ | - | - | - | - |
| | GCN+CCCN | - | - | $2e^{-1}$ | - | $2e^{-4}$ |
| | DGM | $6e^{-4}$ | $4e^{-3}$ | - | - | - |
| | DCM | $1e^{-5}$ | $5e^{-3}$ | $1e^{-1}$ | $2e^{-2}$ | $5e^{-4}$ |
| **Texas** | GCN | $2e^{-5}$ | - | - | - | - |
| | GCN+CCCN | - | - | $1e^{-2}$ | - | $2e^{-4}$ |
| | DGM | $3e^{-5}$ | $9e^{-4}$ | - | - | - |
| | DCM | $1e^{-5}$ | $9e^{-4}$ | $3e^{-2}$ | $7e^{-2}$ | $4e^{-5}$ |

## B  COMPUTATIONAL COMPLEXITY

We carry out a complexity analysis of the proposed architecture with a particular focus on the differences with respect to the DGM (Kazi et al., 2022), which are:

(a) the introduction of $\alpha$-entmax to sample cells compared to the Gumbel-Softmax sampler;

(b) the need to search for all cycles up to length $K_{max}$ in the skeleton $\mathcal{G}_{out}$;

(c) the sampling operation over the cycles;

(d) the cell complex neural network;

For (a), the solution to (3) cannot be expressed in closed form except for specific values of $\alpha$ (Correia et al., 2019), but an $\varepsilon$-approximate solution can be obtained in $\mathcal{O}(1/\log\varepsilon)$ time with a bisection algorithm (Blondel et al., 2019).

For (b), we leverage the algorithm in (Bodnar et al., 2021a), which has complexity $\mathcal{O}((E + N\widetilde{P})\text{polylog}(N))$, where $N$ is the number of vertices, $E$ the number of sampled edges, and $\widetilde{P}$ the number of cycles induced by the skeleton up to length $K_{max}$. Note that in our implementation the cell complex is recomputed for each iteration, but this computation can be amortized across epochs, approximated with stochastic search algorithms, or leveraging more flexible topological spaces, e.g. Combinatorial Complexes.

For (c), the complexity is the same as (a), which is approximately linear in the number $\widetilde{P}$ of cycles.

For (d), the complexity of message-passing is also approximately linear in the size of the cell complex, due to the fact that we consider cells of a constant maximum dimension and boundary size (Bodnar et al., 2021a).

As a further empirical analysis, in Table 3 we break down the actual inference execution times on two reference datasets, Cora and Texas, for GCN, GCN+CCCN, DGM, and DCM, based on the (macro-)operations they require. As we can see from this empirical analysis and as we could expect from the above complexity analysis, the lifting operation (computing the cycles and lifting the node embeddings) and cell sampling (computing the similarities among edge embeddings and applying the $\alpha$-entmax) are the main computational bottlenecks. The lifting operation needs to be performed in every complex-based architecture (we show DCM and GCN+CCCN, but for CWN would be the same). Architectures that do not perform LTI (CWN or GCN+CCCN) can mitigate this problem (only) on transductive tasks by computing the cycles offline. In our setting, additional solutions w.r.t. the ones presented in (b) could be accumulating the gradients and inferring the graph/cell-complexes once every $t$ iteration rather than after every optimization step. The cell sampling bottleneck, unlike the lifting, is not given by any technical requirement, but it is just related to our actual implementation. To keep the code readable and reproducible for (mainly) academic purposes, we use basic data structures and functions, e.g. we store the cells in lists that we parse. However, the bottleneck could be mitigated by optimizing the code. We will soon update our repo.

## C  ADDITIONAL RESULTS

We present a series of additional experiments and ablation studies to further validate the effectiveness of the DCM. In particular, in Section C.1 we investigate the advantages of employing the $\alpha$-entmax as sampling strategy w.r.t. the Gumbel Top-k trick used in the DGM (de Ocáriz Borde et al., 2023;

Table 4: Comparison among sampling strategies on homophilic graph node classification benchmarks. Test accuracy in % avg.ed over 10 splits.

|  |  | Cora | CiteSeer | PubMed | Physics | CS |
|---|---|---|---|---|---|---|
| w/o graph | DCM | $\mathbf{78.80} \pm 1.84$ | $\mathbf{76.47} \pm 2.45$ | $87.38 \pm 0.91$ | $96.45 \pm 0.12$ | $\mathbf{95.40} \pm 0.40$ |
|  | DCM ($\alpha = 1$) | $78.73 \pm 1.99$ | $76.32 \pm 2.75$ | $\mathbf{87.47} \pm 0.77$ | $96.2 \pm 0.27$ | $95.35 \pm 0.37$ |
|  | Top-k DCM | $76.16 \pm 3.71$ | $73.21 \pm 2.73$ | $87.13 \pm 1.22$ | $\mathbf{96.51} \pm 0.49$ | $95.25 \pm 0.08$ |
|  | Top-k DCM All | $76.24 \pm 2.84$ | $73.45 \pm 3.12$ | $87.38 \pm 1.04$ | $96.49 \pm 0.64$ | $95.17 \pm 0.07$ |
| w graph | DCM | $\mathbf{85.78} \pm 1.71$ | $\mathbf{78.72} \pm 2.84$ | $88.49 \pm 0.62$ | $\mathbf{96.99} \pm 0.44$ | $\mathbf{95.79} \pm 0.48$ |
|  | DCM ($\alpha = 1$) | $85.97 \pm 1.86$ | $78.60 \pm 3.16$ | $\mathbf{88.61} \pm 0.69$ | $96.69 \pm 0.46$ | $94.81 \pm 0.49$ |
|  | Top-k DCM | $76.96 \pm 3.46$ | $75.50 \pm 2.15$ | $86.76 \pm 0.89$ | $96.19 \pm 0.33$ | $94.74 \pm 0.28$ |
|  | Top-k DCM All | $77.55 \pm 3.18$ | $75.66 \pm 2.67$ | $86.64 \pm 0.74$ | $96.21 \pm 0.34$ | $94.79 \pm 0.37$ |

Table 5: Comparison among sampling strategies on heterophilic graph node classification benchmarks. Test accuracy in % avg.ed over 10 splits.

|  |  | Texas | Wisconsin | Squirrel | Chameleon |
|---|---|---|---|---|---|
| w/o graph | DCM | $\mathbf{85.71} \pm 7.87$ | $\mathbf{87.49} \pm 5.94$ | $\mathbf{35.55} \pm 2.24$ | $53.63 \pm 3.07$ |
|  | DCM ($\alpha = 1$) | $84.96 \pm 10.24$ | $86.72 \pm 6.02$ | $35.25 \pm 2.22$ | $\mathbf{53.67} \pm 3.19$ |
|  | Top-k DCM | $84.21 \pm 8.21$ | $84.10 \pm 6.60$ | $33.90 \pm 1.38$ | $52.88 \pm 4.24$ |
|  | Top-k DCM All | $82.71 \pm 11.25$ | $85.18 \pm 6.89$ | $33.80 \pm 1.47$ | $53.23 \pm 3.86$ |
| w graph | DCM | $84.87 \pm 10.04$ | $\mathbf{86.33} \pm 5.14$ | $34.95 \pm 2.59$ | $53.05 \pm 3.00$ |
|  | DCM ($\alpha = 1$) | $\mathbf{84.96} \pm 5.60$ | $85.36 \pm 5.05$ | $\mathbf{35.13} \pm 2.27$ | $\mathbf{53.76} \pm 3.72$ |
|  | Top-k DCM | $84.61 \pm 8.47$ | $82.87 \pm 7.59$ | $33.77 \pm 0.91$ | $52.84 \pm 4.24$ |
|  | Top-k DCM All | $84.66 \pm 9.33$ | $82.10 \pm 6.64$ | $33.52 \pm 1.08$ | $51.61 \pm 4.07$ |

Kazi et al., 2022). Additionally, in Section C.2, we study the impact of the number of MP layers in (7)-(4) (implemented, as explained in Section 4 of the body and Appendix F, with GCNs and CCCNs). Finally, in Section C.3, we investigate the usage of different values for the maximum cycle size $K_{max}$.

## C.1 SAMPLING STRATEGIES

In this section, we assess the impact of employing the $\alpha$-entmax as sampling strategy, compared to the Top-k sampler utilized in (Kazi et al., 2022; de Ocáriz Borde et al., 2023), for both homophilic (Table 4) and heterophilic datasets (Table 5), with and without input graph. We compare the two variants of the DCM (again denoted with DCM and DCM ($\alpha = 1$)) against our same architecture but with Gumbel Top-k sampler in place of the $\alpha-$entmax, both when explicit sampling is performed at edge and polygons level (denoted with Top-k DCM) and when only the edge are sampled and all the polygons are taken (the counterpart of DCM ($\alpha = 1$), denoted with Top-K DCM All). Our investigation reveals that the capability of $\alpha$-entmax of generating non-regular topologies leads to significant performance gains on almost all the tested datasets.

## C.2 NUMBER OF MESSAGE-PASSING LAYERS

In this section, we assess the impact of the number of MP layers on the performance of our model. In Table 6 (for homophilic datasets) and Table 7 (for heterophilic datasets), we show the accuracy as a function of the number of MP layers, i.e. GCNs and CCCNs layers at node and edge levels, respectively. We notice that the determination of an optimal number of message-passing layers is not governed by a universal rule but rather depends on the characteristics of the dataset under consideration. However, we observe that employing a single message-passing layer consistently yields favorable performance across most of the datasets, further confirming that integrating higher-order information is beneficial. Moreover, the 1-layer configuration maintains a comparable number of trainable parameters w.r.t. the DGM-M (de Ocáriz Borde et al., 2023) settings, whose results are reported as a comparison. In particular, DGM-M employs 3-layer GCNs while we employ 1-layer GCNs and 1-layer CCCNs (having 3 times the number of parameters of a single GCN layer), see Appendix F for details. For this reason, despite the fact that some results in Table 6 and in Table 7 are better than the ones reported in the body of the paper, we decided to show the ones that correspond

Table 6: Results varying the number of MP layers on homophilic graph node classification benchmarks. Test accuracy in % avg.ed over 10 splits.

| | # MP layers | Cora | CiteSeer | PubMed | Physics | CS |
|---|---|---|---|---|---|---|
| w/o graph | 1 | **78.80** ± 1.84 | **76.47** ± 2.45 | 87.38 ± 0.91 | 96.45 ± 0.12 | **95.40** ± 0.40 |
| | 2 | 76.90 ± 2.81 | 75.86 ± 2.35 | 87.77 ± 0.53 | **97.10** ± 0.15 | 94.81 ± 0.40 |
| | 3 | 74.98 ± 4.20 | 74.59 ± 2.35 | **88.10** ± 0.45 | 97.09 ± 0.11 | 94.15 ± 0.63 |
| | 4 | 69.74 ± 3.95 | 71.41 ± 2.47 | 87.79 ± 0.36 | 96.80 ± 0.08 | 93.54 ± 0.72 |
| w graph | 1 | 85.78 ± 1.71 | **78.72** ± 2.84 | 88.49 ± 0.62 | 96.99 ± 0.44 | **95.79** ± 0.48 |
| | 2 | **88.89** ± 1.63 | 78.42 ± 2.63 | 88.27 ± 0.54 | **97.03** ± 0.23 | 94.50 ± 0.55 |
| | 3 | 88.07 ± 2.26 | 76.95 ± 2.84 | **88.97** ± 0.61 | 97.01 ± 0.12 | 94.47 ± 0.54 |
| | 4 | 86.00 ± 2.21 | 76.62 ± 2.86 | 88.62 ± 0.76 | 96.86 ± 0.19 | 93.66 ± 0.51 |

to architectures whose number of trainable parameters are as similar as possible to the reported competitors.

### C.3 MAXIMUM CYCLE SIZE

In this section, we conduct an ablation study on the maximum length $K_{max}$ of induced cycles taken in consideration to sample the polygons of the latent cell complex. In Table 8 and in Table 9, we show the accuracy as a function of $K_{max}$, respectively. As for the choice of the number of MP layers, even the optimal $K_{max}$ varies based on the specific dataset, however with top performance obtained in most cases when $K_{max} = 4$.

## D ALGEBRAIC DESCRIPTION OF REGULAR CELL COMPLEXES

It is possible to give a combinatorial description along with a rich algebraic representation of regular cell complexes. To do so, it is essential to introduce an orientation of the cells. Orienting cells is not mathematically trivial but, in the end, it is only a "bookkeeping matter" (Roddenberry et al., 2022). One of the possible ways of orienting cells (Sardellitti et al., 2021) is via a simplicial decomposition of the complex, i.e. subdividing the cell into a set of internal $k$-simplices (Lee, 2000; Barbarossa & Sardellitti, 2020), so that i) two simplices share exactly one $(k-1)$-simplicial boundary element, which is not the boundary of any other $k$-cell in the complex; and ii) two $k$-simplices induce an opposite orientation on the shared $(k-1)$-boundary. Therefore, by orienting a single internal simplex, the orientation propagates on the entire cell.

Given an orientation, the structure of an oriented regular cell complex of order $K$ is then fully captured by the set of its incidence (or boundary) matrices $\mathbf{B}_k \in \mathbb{R}^{N_{k-1} \times N_k}$, $k = 1, \ldots, K$, with entries $\mathbf{B}_k(i, j) = 0$ if $\sigma_i^{k-1}$ is not a face of $\sigma_j^k$, and $\mathbf{B}_k(i, j) = 1$ (or $-1$), if $\sigma_i^{k-1}$ is a face of $\sigma_j^k$ and its orientation is coherent (or not) with the orientation of $\sigma_j^k$. From the incidence information, we build the Hodge (or combinatorial) Laplacian matrices of order $k = 0, \ldots, K$ as (Goldberg, 2002):

$$\mathbf{L}_0 = \mathbf{B}_1 \mathbf{B}_1^T, \tag{14}$$

$$\mathbf{L}_k = \underbrace{\mathbf{B}_k^T \mathbf{B}_k}_{\mathbf{L}_k^{(d)}} + \underbrace{\mathbf{B}_{k+1} \mathbf{B}_{k+1}^T}_{\mathbf{L}_k^{(u)}}, \quad k = 1, \ldots, K-1, \tag{15}$$

$$\mathbf{L}_K = \mathbf{B}_K^T \mathbf{B}_K. \tag{16}$$

Table 7: Results varying the number of MP layers on heterophilic graph node classification benchmarks. Test accuracy in % avg.ed over 10 splits.

| | # MP layers | Texas | Wisconsin | Squirrel | Chameleon |
|---|---|---|---|---|---|
| w/o graph | 1 | 85.71 ± 7.87 | **87.49** ± 5.94 | **35.55** ± 2.24 | **53.63** ± 3.07 |
| | 2 | **87.89** ± 10.24 | 84.80 ± 4.81 | 33.30 ± 1.67 | 52.22 ± 4.22 |
| | 3 | 82.02 ± 8.97 | 83.64 ± 4.87 | 33.61 ± 2.55 | 51.34 ± 3.25 |
| | 4 | 80.75 ± 10.80 | 79.49 ± 7.45 | 31.96 ± 2.56 | 47.82 ± 3.11 |
| w graph | 1 | **84.87** ± 10.04 | **86.33** ± 5.14 | **34.95** ± 2.59 | **53.05** ± 3.00 |
| | 2 | 77.34 ± 8.64 | 76.31 ± 4.92 | 33.53 ± 2.24 | 52.61 ± 2.73 |
| | 3 | 77.34 ± 8.09 | 80.95 ± 6.52 | 33.18 ± 1.88 | 51.47 ± 2.80 |
| | 4 | 74.76 ± 12.01 | 76.13 ± 7.43 | 33.45 ± 2.59 | 43.65 ± 3.07 |

Table 8: Results varying $K_{max}$ on homophilic graph node classification benchmarks. Test accuracy in % avg.ed over 10 splits.

| | $K_{max}$ | Cora | CiteSeer | PubMed | Physics | CS |
|---|---|---|---|---|---|---|
| w/o graph | 3 | $78.38 \pm 2.50$ | $75.56 \pm 1.45$ | $87.44 \pm 0.91$ | $96.32 \pm 0.28$ | $\mathbf{95.41} \pm 0.60$ |
| | 4 | $\mathbf{78.80} \pm 1.84$ | $\mathbf{76.47} \pm 2.45$ | $87.38 \pm 0.91$ | $96.45 \pm 0.12$ | $95.40 \pm 0.40$ |
| | 5 | $78.60 \pm 2.60$ | $75.62 \pm 1.26$ | $\mathbf{87.54} \pm 0.98$ | $96.20 \pm 0.28$ | $95.39 \pm 0.60$ |
| w graph | 3 | $85.54 \pm 2.40$ | $78.38 \pm 1.56$ | $\mathbf{88.50} \pm 0.78$ | $97.01 \pm 0.28$ | $95.68 \pm 0.63$ |
| | 4 | $\mathbf{85.78} \pm 1.71$ | $\mathbf{78.72} \pm 2.84$ | $88.49 \pm 0.62$ | $96.99 \pm 0.44$ | $\mathbf{95.79} \pm 0.48$ |
| | 5 | $85.54 \pm 2.43$ | $78.32 \pm 1.47$ | $85.75 \pm 1.02$ | $\mathbf{97.07} \pm 0.40$ | $95.68 \pm 0.56$ |

Table 9: Results varying $K_{max}$ on heterophilic graph node classification benchmarks. Test accuracy in % avg.ed over 10 splits.

| | $K_{max}$ | Texas | Wisconsin | Squirrel | Chameleon |
|---|---|---|---|---|---|
| w/o graph | 3 | $85.53 \pm 9.85$ | $84.80 \pm 6.29$ | $35.15 \pm 1.76$ | $53.41 \pm 3.30$ |
| | 4 | $\mathbf{85.71} \pm 7.87$ | $\mathbf{87.49} \pm 5.94$ | $\mathbf{35.55} \pm 2.24$ | $\mathbf{53.63} \pm 3.07$ |
| | 5 | $80.22 \pm 11.54$ | $85.57 \pm 7.19$ | $34.98 \pm 2.15$ | $53.01 \pm 3.73$ |
| w graph | 3 | $\mathbf{85.49} \pm 7.89$ | $80.95 \pm 7.36$ | $33.11 \pm 2.58$ | $52.70 \pm 3.21$ |
| | 4 | $84.87 \pm 10.04$ | $\mathbf{86.33} \pm 5.14$ | $\mathbf{34.95} \pm 2.59$ | $\mathbf{53.05} \pm 3.00$ |
| | 5 | $84.21 \pm 9.15$ | $81.71 \pm 6.33$ | $33.88 \pm 1.49$ | $52.92 \pm 3.30$ |

All Laplacian matrices of intermediate orders, i.e., $k = 1, \ldots, K - 1$, contain two terms: The first term $\mathbf{L}_k^{(d)}$, also known as lower Laplacian, encodes the lower adjacency among $k$-order cells, i.e. $\mathbf{L}_k^{(d)}(i,j) = 0$ if $\sigma_j^k \notin \mathcal{N}_d(\sigma_i^k)$; the second term $\mathbf{L}_k^{(u)}$, also known as upper Laplacian, encodes the upper adjacency among $k$-order cells, i.e. $\mathbf{L}_k^{(u)}(i,j) = 0$ if $\sigma_j^k \notin \mathcal{N}_u(\sigma_i^k)$. $\mathbf{L}_0$ is the usual graph Laplacian. Hodge Laplacians admit the following Hodge decomposition (Lim, 2020).

***Proposition 1 (Hodge Decomposition)*** The $k$-topological signal space $\mathbb{R}^{N_k}$ can be decomposed as (Grady & Polimeni, 2010):

$$\mathbb{R}^{N_k} = \mathbf{im}\big(\mathbf{B}_k^T\big) \bigoplus \mathbf{im}\big(\mathbf{B}_{k+1}\big) \bigoplus \mathbf{ker}\big(\mathbf{L}_k\big), \tag{17}$$

where $\bigoplus$ is the direct sum of vector spaces, and $\mathbf{ker}(\cdot)$ and $\mathbf{im}(\cdot)$ are the kernel and image spaces of a matrix, respectively.

Let us denote the $i$-th column of any feature matrix $\mathbf{X}_k \in \mathbb{R}^{N_k \times F}$, i.e. the vector collecting the $i$-th signal values of all the cells, with $\mathbf{x}_k[i] := [x_{k,i}(\sigma_1^k), \ldots, x_{k,i}(\sigma_{N_k}^k)] \in \mathbb{R}^{N_k}$, where $x_{k,i}$ is defined as in (1).

$k$-cell signals can be represented over the basis of the eigenvectors of the corresponding Hodge Laplacian matrix $\mathbf{L}_k$. We denote the eigendecomposition of $\mathbf{L}_k$ as $\mathbf{L}_k = \mathbf{U}_k \mathbf{\Lambda}_k \mathbf{U}_k^T$, where $\mathbf{U}_k$ and $\mathbf{\Lambda}_k = \mathrm{diag}\{\lambda_{k,1}, \ldots, \lambda_{k,N_k}\}$ collect the eigenvector and the eigenvalues of $\mathbf{L}_k$, respectively. The Cell Complex Fourier Transform (CFT) of order $k$ is defined as the projection of a $k$-cell signal $\mathbf{x}_k[i]$ onto the eigenvectors of $\mathbf{L}_k$ (Sardellitti et al., 2021):

$$\widehat{\mathbf{x}}_k[i] \triangleq \mathbf{U}_k^T \mathbf{x}_k[i]. \tag{18}$$

We refer to the eigenvalue domain of the CFT as the frequency domain (or spectrum). A consequence of the Hodge decomposition in (17) is that the eigenvectors belonging to $\mathbf{im}\big(\mathbf{L}_k^{(d)}\big)$ are orthogonal to those belonging to $\mathbf{im}\big(\mathbf{L}_k^{(u)}\big)$, for all $k = 1, \ldots, K - 1$.

## E   ADDITIONAL PROPERTIES AND PLOTS OF THE INFERRED COMPLEXES

In this appendix, we analyze properties and show additional plots of the inferred latent cell complexes for the Wisconsin and Cora datasets. The plots shown in Figure 5 for the Wisconsin dataset, as the ones shown in Figure 4, are obtained by training the architecture without providing the input graph. The plots in Figure 6 for the Cora dataset show the input and the final inferred latent complexes both when the input graph is provided or not to the DCM. Overall, Figure 5 and Figure 6 further show the ability of the DCM to learn a latent complex with a high level of homophily. Moreover, in Tables 11 and 12 we show the average homophily level of the learned graphs for all the tested datasets, again both when the input graph is provided or not. It is worth noticing that the homophily levels of the learned graphs are always greater than the input graph, even on the homophilic datasets.

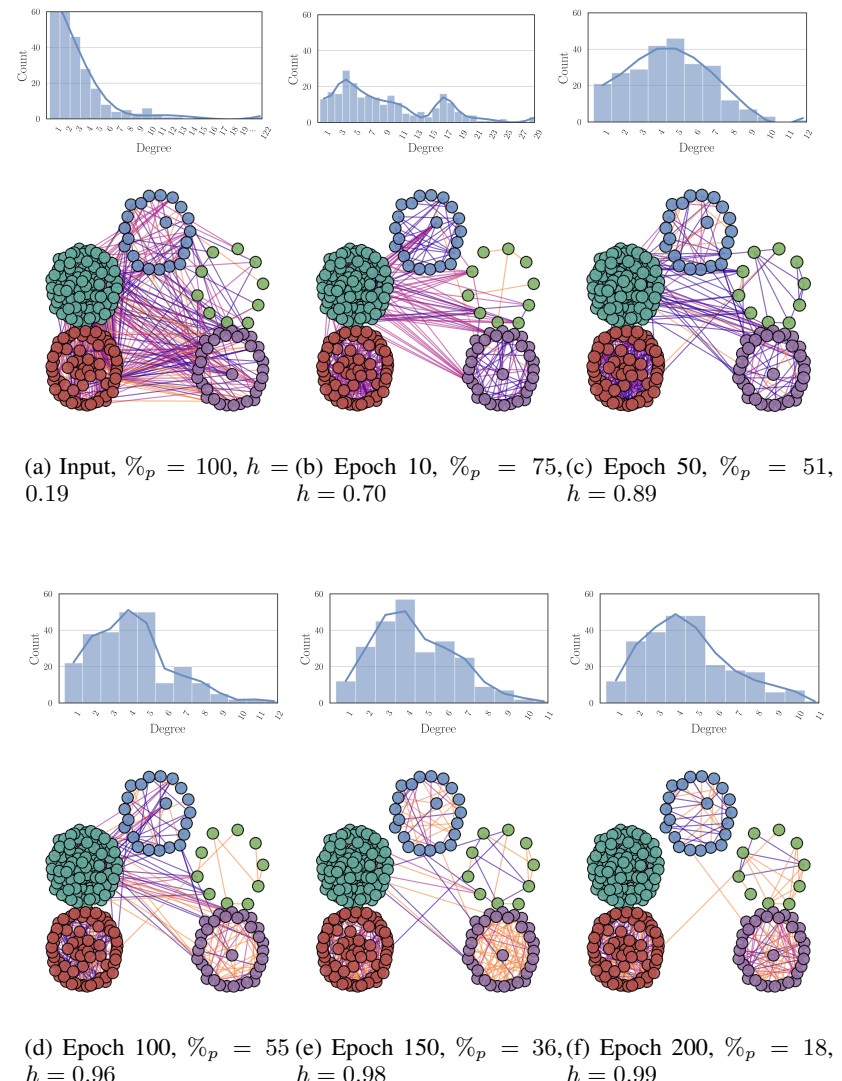

(a) Input, $\%_p = 100$, $h =$ (b) Epoch 10, $\%_p = 75$, (c) Epoch 50, $\%_p = 51$,
0.19                  $h = 0.70$              $h = 0.89$

(d) Epoch 100, $\%_p = 55$ (e) Epoch 150, $\%_p = 36$, (f) Epoch 200, $\%_p = 18$,
$h = 0.96$                $h = 0.98$             $h = 0.99$

Figure 5: Evolution of the latent complex for the Wisconsin dataset, along with homophily level and nodes degree distribution. Edges in orange, triangles in lilac, squares in purple ($K_{max} = 4$)

To give some quantitative hints of the interpretability of the inferred cell complexes, we now present several graph and complex-based metrics for the Cora and Wisconsin datasets, which illustrate how the inferred cell complexes align with amenable properties that would be intuitively expected. In particular, we assess the properties of the complexes at various levels.

**Topological properties.** To assess the latent topology w.r.t. the input graphs of the considered datasets, we evaluate the normalized total spectral gap between the input graph and the inferred graph, i.e. the sum of differences between the eigenvalues of the input and inferred graph Laplacians divided by the number of nodes. Two graphs can be considered topologically similar if the spectral gap is small (Dong et al., 2020), i.e. if they have similar frequencies, in the sense of Appendix D. In particular, we first orient the input complex $\mathcal{C}_{\mathcal{G}_{in}}$ (i.e. the complex having the input graph $\mathcal{G}_{in}$ as 1-skeleton and all of its cycles as polygons) and the inferred latent complex $\mathcal{C}_{\mathcal{G}_{out}}$ using the procedure presented in Appendix D. At this point, we compute the graph Laplacians $\mathbf{L}_{0,in}$ and $\mathbf{L}_{0,out}$ of the input $\mathcal{G}_{in}$ and inferred $\mathcal{G}_{out}$ graphs, respectively. We compute their eigenvalues $\{\lambda_{0,i,in}\}_i$ and $\{\lambda_{0,i,out}\}_i$, and we evaluate their normalized total spectral gap $NSG(\mathcal{G}_{in}, \mathcal{G}_{out})$, defined as:

$$NSG(\mathcal{G}_{in}, \mathcal{G}_{out}) = \frac{1}{N} \sum_{i=1}^{N} (\lambda_{0,i,in} - \lambda_{0,i,out})^2 \tag{19}$$

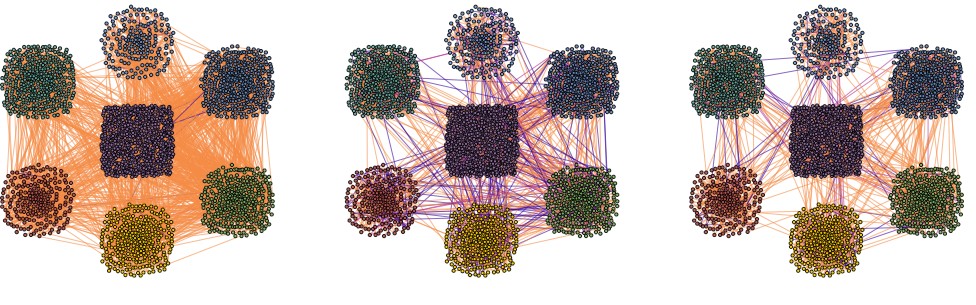

(a) Input graph      (b) $\mathcal{C}_{\mathcal{G}_{out}}$ w input graph, $h = 0.89$   (c) $\mathcal{C}_{\mathcal{G}_{out}}$ w/o input graph, $h = 0.93$

Figure 6: Input graph (a), inferred latent complex if the input graph is provided (b), and inferred latent complex if the input graph is provided (c) for the Cora dataset, along with homophily level. Edges in orange, triangles in lilac, squares in purple ($K_{max} = 4$).

In Table 10, we show this metric when the input graph is provided or not. As the reviewer can notice, the spectral gap on the Cora dataset is in both cases an order of magnitude smaller than the spectral gap on the Wisconsin dataset. This result allows us to state that, if the input graph is provided, our model is able to significantly discard information about it in case it is a "bad" graph (Wisconsin, heterophilic), while it is able to retain its main topological properties and sufficiently rewire it to improve the downstream task if it is a "good" graph (Cora, homophilic). On the other hand, if the input graph is not provided, our model is able to infer a topologically similar graph to the (unseen) input graph if the input graph is "good" (Cora), while it infers a topologically different graph if the input graph is "bad" (Wisconsin).

**Homophily properties.** To assess the latent topology w.r.t. the node labels $\mathbf{y} \in \mathbb{R}^N$ of the considered datasets, we employ the Polygon Homophily, which quantifies the fraction of homogeneous polygons that connect nodes having the same label, respectively. In particular, the Polygon Homophily is designed by us by generalizing the simplicial homophily from (Sarker et al., 2023), and for a regular cell complex $\mathcal{C}_{\mathcal{G}}$ as the ones we consider, it is computed as:

$$H(\mathbf{y}, \mathcal{C}_{\mathcal{G}}) = \frac{|\mathcal{P}_h| \cdot |\widetilde{\mathcal{P}}_h|}{|\mathcal{P}| \cdot |\widetilde{\mathcal{P}}|}, \tag{20}$$

where $\mathcal{P}$ is the set of polygons of the complex $\mathcal{C}_{\mathcal{G}}$, $\widetilde{\mathcal{P}}$ is the set of induced cycles of the 1-skeleton (graph) $\mathcal{G}$, $\mathcal{P}_h \subseteq$ is the subset of homogeneous polygons, i.e. polygons whose vertices have the same label, and $\widetilde{\mathcal{P}}_h$ is the subset of homogeneous induced cycles. The polygon homophily reduces to the usual (graph) homophily if edges are considered in place of polygons. With these definitions, a complex exhibits high polygon homophily if $H(\mathbf{y}, \mathcal{C}_{\mathcal{G}}) \geq 1$. In Table 10, we show the polygon homophily $H(\mathbf{y}, \mathcal{C}_{\mathcal{G}_{out}})$ of the inferred complex $\mathcal{C}_{\mathcal{G}_{out}}$, when the input graph is provided or not. As the reader can notice combining Table 11, Table 12, and Table 10, the inferred complexes show both graph and polygon homophily.

**Feature-dependent properties.** To evaluate how the latent topology evolves w.r.t. node features on the considered datasets, we use the notion of total variation (TV) of features defined over nodes and edges (Sardellitti et al., 2021). In particular, the total variation $TV_0$ at the node level is a measure of the smoothness of node features, i.e. if node features of nodes linked by an edge are similar, the node TV will be small; the total variation $TV_1$ at the edge level is a measure of the smoothness of edge features, i.e. if edge features of edges belonging to the same polygon are similar, the edge TV will be small. Given an orientated cell complex $\mathcal{C}_{\mathcal{G}}$, its order 0 (graph) $\mathbf{L}_0$ and order 1 $\mathbf{L}_1$ Laplacians, node and edge feature matrices $\mathbf{X}_0$ and $\mathbf{X}_1$, respectively, then the total variation at node and edge level can be computed as:

$$TV_0(\mathbf{X}_0, \mathcal{C}_{\mathcal{G}}) = \text{Trace}\{\mathbf{X}_0^T \mathbf{L}_0 \mathbf{X}_0\}, \tag{21}$$

$$TV_1(\mathbf{X}_1, \mathcal{C}_{\mathcal{G}}) = \text{Trace}\{\mathbf{X}_1^T \mathbf{L}_1^{(u)} \mathbf{X}_1\}. \tag{22}$$

In other words, $TV_0$ is the sum of the squared variations of the node signals along the edge of the complex, while $TV_1$ is the sum of the squared variations (curls) of the edge signals along the polygons

of the complex. In Table 10, we show the TV $TV_0(\mathbf{X}_{0,in}, \mathcal{C}_{\mathcal{G}_{in}})$ of the initial node features over the input graph, the TV $TV_0(\mathbf{X}_{0,in}, \mathcal{C}_{\mathcal{G}_{out}})$ of the initial node features over the inferred graph, the TV $TV_0(\mathbf{X}_{0,out}, \mathcal{C}_{\mathcal{G}_{in}})$ of the final node features over the input graph, the TV $TV_0(\mathbf{X}_{0,out}, \mathcal{C}_{\mathcal{G}_{out}})$ of the final node features over the inferred graph, the TV $TV_1(\mathbf{X}_{1,out}, \mathcal{C}_{\mathcal{G}_{in}})$ of the final edge features over the input complex (obtained again considering all the induced cycles as polygons), and the TV $TV_1(\mathbf{X}_{1,out}, \mathcal{C}_{\mathcal{G}_{out}})$ of the final edge features over the inferred complex. As the reader can notice, the DCM is always able to learn embeddings that are smooth over the inferred complex, both at node and edge level, when the input graph is provided or not. Interestingly, not only the learned embeddings are smooth, but also the input features are smoother on the 1-skeleton (graph) of the inferred complex than on the input graph.

The above quantitative analysis, along with the qualitative information given by the various plots, show that the DCM is able to learn complexes that jointly align with data features and data labels, while at the same time being able to extract or retain topological information similar to the input graph if it is useful for downstream task. These facts could be particularly useful in more involved and tailored applications, and we plan to investigate this direction in future (more) applied works.

## F  MODEL ARCHITECTURE

In this appendix, we present a detailed description of the architectures employed to obtain the results in Table 1 and Table 2. To ensure uniformity and show the performance gain without intensive ad-hoc hyperparameters tuning (as the ones performed for the DGM (Kazi et al., 2022) and the DGM-M (de Ocáriz Borde et al., 2023)), we maintained a constant configuration for the number of layers, hidden dimensions, activation functions, $K_{max}$ (4), (pseudo-)similarity functions (minus the euclidean distances among embeddings), dropout rates (0.5), and learning rates (0.01) across all datasets. The architecture details are shown in Tables 13 and 14. We conducted training for a total of 200 epochs for the homophilic datasets, with the exception of the physics dataset, which underwent 100 epochs like the heterophilic datasets. Our experiments were performed using a single NVIDIA RTX A6000 with 48 GB of GDDR6 memory. As mentioned in Section 4, in every experiment we employ Graph Convolutional Neural Networks (GCNs) and Cell Complex Convolutional Neural Network (CCCNs) as specific MP architectures at node and edge levels, respectively. We give a brief description of GCNs and CCCNs in the following.

**Remark 5.** The DCM has not to be considered a novel GNN architecture that aims to achieve SOTA performance on graph benchmarks. To the best of our knowledge, DCM is the first model that permits learning of higher-order latent interactions while overcoming many of the LGI's (and TDL's) limitations and achieving and matching SOTA results across multiple datasets. In addition, please note that the majority of GNN and TNN architectures could be integrated into DCM as backbones at the node and edge levels as instead of the employed GCN and CCCN.

### F.1  GRAPH AND CELL COMPLEX CONVOLUTIONAL NEURAL NETWORKS

**Graph Convolutional Neural Networks** (GCNs) are one of the most famous and simple GNNs architectures. In GCNs, the output features of a node are computed as a weighted sum of linearly transformed features of its neighboring nodes. Therefore, the MP round in (4) is implemented as:

$$\mathbf{x}_{0,int}(i) = \gamma_0 \left( \sum_{j \in \mathcal{N}_u(\sigma_i^0)} a_{i,j} \mathbf{x}_{0,in}(j) \mathbf{W} \right), \tag{23}$$

where $\mathbf{W}$ are learnable parameters and the (normalized) weights $a_{i,j}$ are set as $a_{i,j} = \frac{1}{\sqrt{d_j d_i}}$, with $d_i$ and $d_j$ being the degrees of node $i$ and node $j$, respectively (Kipf & Welling, 2017a).

Table 10: LTI metrics.

| | | $TV_0(\mathbf{X}_{0,in}, \mathcal{C}_{\mathcal{G}_{out}})$ | $TV_0(\mathbf{X}_{0,in}, \mathcal{C}_{\mathcal{G}_{in}})$ | $TV_0(\mathbf{X}_{0,out}, \mathcal{C}_{\mathcal{G}_{out}})$ | $TV_0(\mathbf{X}_{0,out}, \mathcal{C}_{\mathcal{G}_{in}})$ | $TV_1(\mathbf{X}_{1,out}, \mathcal{C}_{\mathcal{G}_{out}})$ | $TV_1(\mathbf{X}_{1,out}, \mathcal{C}_{\mathcal{G}_{in}})$ | $H(\mathbf{y}, \mathcal{C}_{\mathcal{G}_{out}})$ | $NSG(\mathcal{G}_{in}, \mathcal{G}_{out})$ |
|---|---|---|---|---|---|---|---|---|---|
| cora | Graph | 5736 | 8786 | 457 | 1912 | 538 | 871 | .95 | $3.4 \cdot 10^{-4}$ |
| | w/o Graph | 5623 | 8786 | 316 | 2651 | 461 | 502 | .99 | $1.8 \cdot 10^{-4}$ |
| wis. | Graph | 853 | 1201 | 396 | 1883 | 329 | 1087 | .75 | $5.5 \cdot 10^{-3}$ |
| | w/o Graph | 888 | 1201 | 28 | 1783 | 258 | 1299 | .94 | $7.2 \cdot 10^{-3}$ |

Table 11: Homophily level of the latent graph on homophilic graph node classification benchmarks.

|  | Cora | CiteSeer | PubMed | Physics | CS |
|---|---|---|---|---|---|
| w/o graph | 0.93 | 0.96 | 0.84 | 0.98 | 0.99 |
| w graph | 0.89 | 0.78 | 0.82 | 0.96 | 0.92 |
| input | 0.81 | 0.74 | 0.80 | 0.93 | 0.80 |

Table 12: Homophily level of the latent graph on heterophilic graph node classification benchmarks.

|  | Texas | Wisconsin | Squirrel | Chameleon |
|---|---|---|---|---|
| w/o graph | 0.99 | 0.99 | 0.30 | 0.64 |
| w graph | 0.80 | 0.70 | 0.29 | 0.62 |
| input | 0.11 | 0.21 | 0.22 | 0.23 |

**Cell Complex Convolutional Neural Networks** (CCCNs) generalize GCNs to cell complexes using the adjacencies introduced in Definition 1. In our case, the output features of an edge are computed as a weighted sum of linearly transformed features of its neighboring edges, over the upper and lower adjacencies. Therefore, in this paper, we implement the MP round in (7) as:

$$\mathbf{x}_{1,out}(i) = \gamma_1 \left( \sum_{j \in \mathcal{N}_u(\sigma_i^1)} a_{u,i,j} \mathbf{x}_{1,int}(j) \mathbf{W}_u + \sum_{j \in \mathcal{N}_d(\sigma_i^1)} a_{d,i,j} \mathbf{x}_{1,int}(j) \mathbf{W}_d + \mathbf{x}_{1,int}(j) \mathbf{W} \right),$$
(24)

where $\mathbf{W}_u$, $\mathbf{W}_d$, and $\mathbf{W}$ are learnable parameters. The weights are normalized with the same approach of GCNs, with upper (for the $a_{u,i,j}$s) and lower (for the $a_{d,i,j}$s) degrees. The skip connection $\mathbf{x}_{1,int}(j)\mathbf{W}$ is as usual beneficial, and in the case of CCCNs it has a further theoretical justification in terms of Hodge Decomposition and signal filtering, see (Roddenberry et al., 2022; Sardellitti et al., 2021) for further details.

Table 13: Model Architecture.

| No. Layer param. | Activation | LayerType |
|---|---|---|
| (no. input features, 32) | ReLU | Linear |
|  |  | DCM |
| (32, 32) | ReLU | Graph Conv |
| (32, 32) | ReLU | Cell Conv |
| (64, no. classes) | Softmax | Linear |

Table 14: DCM Architecture.

|  |  | DCM* | DCM |
|---|---|---|---|
| No. Layer param. | Activation | Layer type | |
| (32, 32) | ReLU | Linear | Graph Conv |
| (32, 32) | ReLU | Linear | Graph Conv |
| (32, 32) | None | Linear | Graph Conv |

