# OpenReview forum: "From Latent Graph to Latent Topology Inference: Differentiable Cell Complex Module"
_ICLR.cc/2024/Conference — ICLR 2024 poster_

### Official Review · Reviewer_EucF · 2023-10-19

**Soundness:** 3 good
**Presentation:** 2 fair
**Contribution:** 3 good
**Rating:** 8
**Confidence:** 4

**Summary:**

The authors introduce a novel Latent Topology Inference (LTI) method that enables the learning of non-regular topologies based on their novel Differentiable Cell Complex Module (DCM). DCM is designed to compute cell probabilities within the complex, thus enhancing downstream tasks. They show how to integrate DCM with cell complex message-passing network layers and train it in an end-to-end fashion, offering significant improvements, especially in cases where an input graph is not provided. The paper demonstrates the effectiveness of the proposed approach on both homophilic and heterophilic graph datasets.

**Strengths:**

- The novel $\alpha$-DGM demonstrates remarkable efficacy in characterizing the 1-skeleton of the latent cell complex.
- The paper substantiates its claims with extensive experimental results, both in the main body and the appendix, providing a thorough evaluation of the proposed method.
- The "Limitations" section is thoughtfully composed, addressing potential constraints and challenges of the approach.

**Weaknesses:**

- Section 3.1 initially introduces the $\alpha$-DGM; however, the subsequent description within this section appears to be inconsistent with the concept of $\alpha$-DGM.
- Section 3 would benefit from a reorganization to enhance clarity and coherence. Its current form significantly hinders the understanding of the proposed method. Please reorganize it for readers to follow the description.
- Notably, there is no dedicated "Reproducibility Statement" section in the paper, which hinders providing clear instructions for reproducing the results. The inclusion of such a section would enhance the paper's accessibility and reproducibility.

**Questions:**

- Same as I stated in the section of weakness, on of my major concerns it the organization of Section 3. More efforts are needed in this part. The following few questions might help the authors to refine the section.
- In the paragraph above 'Remark 2', the layer normalization $\mathcal{LN}$ is employed. If I understand correctly, $\mathcal{LN}$ actually works on vectors of similarities. Thus is it really precise to call it as layer normalization?
- In the same paragraph, the threshold of assigning an edge is set as $0$. Is there any ablation study on the selection of this threshold?
- In the same paragraph, what is 'parameter $\alpha$'? Is it mentioned before?
- In Eqn. (5), how to get $\mathcal{B}(\sigma_i^1)$?
- Similarly, in Eqn. (9),  how to get $\mathcal{CB}(\sigma_i^0)$?
- As shown in Table 2, the performance of DCM is higher without an initial graph than with an initial graph. What could be the possible reason behind this observation?
- How to modify the proposed method to make it work for directed complexes?

I will raise my score if my concerns are correctly addressed.

---

> ### Author Response · Authors · 2023-11-17
>
> ## Weaknesses
>
> 1. In our paper, DGM refers to the original Differentiable Graph Module of Kazi et al. (2022). Section 3.1 introduces $\alpha$-DGM, which is our prosed variation of DGM that exploits the entmax function instead of Gumbel-Softmax to sample a potentially irregular graph topology. $\alpha$-DGM is the variant we use in our Differentiable Cell Module (DCM), as shown e.g. in Figure 3. This is also reflected in the experiments, where we use DGM only as a baseline model to refer to the original variant of Kazi et al. (2022). We clarified this point in the revised version of the paper.
>
> 2. We modified the Section based on the reviewers’ suggestions (please also see our other answers). The organization is as follows: we provide a general overview of the method (Fig. 2), before describing each component in turn, (i) the graph sampling procedure (Section 3.1), the lifting procedure (Section 3.2), our novel polygon sampling method (Section 3.3, Fig. 3), and the final downlifting procedure (Section 3.4). Reviewer XizW suggested this to be the most natural exposition, however, we would be glad to do additional rewriting or reorganization if the reviewer has further suggestions.
>
> 3. We added a reproducibility statement immediately after the references and before Appendix A. Please note that we provide a link to an anonymous repository with code allowing to reproduce our experiments.
>
> ## Questions
>
> 1. See Weaknesses 1-2.
>
> 2. At the graph level, this operation is applied to a matrix containing row-wise the probability vectors $\mathbf{p}_i$. Each row is normalized based on its empirically computed mean and variance. Hence, this is equivalent to LN as generically used for sequences or sets in transformers. This is also matched in the code (see our previous answer), where we applied the default LN implementation from PyTorch.
>
> 3. We set it to $0$ because a similar effect to modifying the threshold can be obtained in the $\alpha$-DGM module by varying the $\alpha$ parameter, which is optimized by gradient descent and does not require manual fine-tuning (which may be time-consuming and dataset-dependent). Nevertheless, we tested two additional values for the threshold (0.1 and 0.01) on a subset of the datasets obtaining similar results, as shown in the following Table.
>
> |                   |          | Th = 0   | Th = 0.01 | Th = 0.1  |
> |-------------------|----------|----------|-----------|-----------|
> | **Citeseer**      | w graph  | 78.72%   | 78.88%    | 78.54%    |
> |                   | w/o graph| 76.47%   | 76.16%    | 76.32%    |
> | **Cora**          | w graph  | 85.78%   | 86.33%    | 85.41%    |
> |                   | w/o graph| 78.80%   | 78.77%    | 78.47%    |
> | **Texas**         | w graph  | 84.87%   | 87.13%    | 83.60%    |
> |                   | w/o graph| 85.71%   | 85.48%    | 84.65%    |
> | **Wisconsin**     | w graph  | 86.33%   | 86.10%    | 86.11%    |
> |                   | w/o graph| 87.49%   | 85.95%    | 87.10%    |
>
>
> 4. We describe the meaning and the usage of $\alpha$ below Eq. (5). In the revised paper, we made it clear that the Tsallis $\alpha$-entropy is formally a class of functions parametrized by a parameter $\alpha \geq 1$.
>
> 5. The boundary of an edge $i$ is simply made by its endpoints node $j$ and $v$. We clarified it below Eq. (5) in the revised paper.
>
> 6. The coboundary of a node $i$ is simply made by all the edges for which node $i$ is an endpoint. We clarified it below Eq. (9) in the revised paper.
>
> 7. Table 2 is about heterophilic datasets. We expect the DCM to achieve better performance when the graph is not provided. This is because, in the heterophilic setting, the input graph and data structure are in contradiction with the homophily assumption on which most of the well-known Graph and Topological NNs hinge, i.e. connected nodes have similar labels. Indeed, our method achieves top performance without an input graph for the heterophilic datasets and with an input graph for the homophilic datasets, as one could expect. However, even observing the results across both homophilic and heterophilic datasets when the input graph is provided, we can appreciate how DCM exploits the available "good'' graphs in the homophilic case while being less sensitive to the "wrong'' ones in the heterophilic case.
>
> 8. Implementing Latent Topology Inference for directed complexes is one of the future directions we are definitely most interested in. However, it is not trivial, because fundamental literature about how to theoretically characterize directed regular cell complexes is missing. Few works analyzed directed simplicial complexes (very loosely speaking, a particular case of cell complexes), however, they could be pretty limited as objects to leverage in the context of Latent Topology Inference.

---

> > ### Comment · Reviewer_EucF · 2023-11-20
> >
> > Dear Authors,
> >
> > Thank you for your rebuttal and the revisions made to the paper. Your responses have effectively addressed my concerns, leading me to consider a higher review score.
> >
> > Cheers,
> >
> > Reviewer EucF

---

### Official Review · Reviewer_XizW · 2023-10-31

**Soundness:** 4 excellent
**Presentation:** 3 good
**Contribution:** 3 good
**Rating:** 8
**Confidence:** 3

**Summary:**

This paper proposes a method to learn "latent topology" in ML tasks, extending Latent Graph Inference. The idea is to learn a cell complex during training. Their method extends the Differentiable Graph Module (DGM) (Kazi 2022) to Diff. Cell Module (DCM). They also use $\alpha$-maxent instead of gumball softmax. As I understand it, the procedure involves:
1. Get auxiliary node features $x_{0,aux}= \nu (x_{0,in})$ from inputs $x_{0,in}$, (via GNN, if a graph is given, or MLP otherwise).
2. Use $\alpha$-maxent to infer a graph (1-simplex).
3. Build higher order cells (e.g. faces, volumes etc) using message passing (MP) on the inferred lower-order cells (e.g. edges)
4. Do MP for inference using cell complex conv nets (CCCN).

In the paper, they mostly only discuss up to 2-simplexes, with the Polygon Inference Module (PIM), and not higher simplexes.

They conduct a set of experiments on the usual graph benchmarks and show superior performance against a few baselines, including DGM. Most notably, they point out that most graph based methods (if the graph is similarity based) do not perform well on heterophilic datasets. They find that their method (using 2-simplexes) usually outperforms other methods.

**Strengths:**

1. The idea of inferring cell complexes is a nice and natural extension of learning graphs.
2. Although the number of higher-order cells can grow and quickly become intractable, they seem to use methods that makes this inference manageable (Appendix B and sec. 3.3).
3. It outperforms others on heterophilic datasets (Table 2)
4. Extensive tests and ablation studies.
5. Well-written with detailed appendix and discussion of limitations

**Weaknesses:**

1. In PIM, eq (6) restricts the types of polygons that can be inferred (see questions)
2. Limiting PIM to small polygons may fail to capture long-range dependencies
3. When a graph is given, the experiments are inconclusive. I appreciate reporting the negative results. But a discussion of the failure cases would be beneficial.

**Questions:**

1. How efficient is PIM in practice? How does the training time compare with DGM or other baselines? I know you discuss the time complexity in App B, but I'm curious about the wall-clock time.
2. For eq (6), is there a reasoning saying higher order correlations would be weaker, e.g. randomness of $x_{1,int}$? In principle, you could also consider any contraction, including products of all three $ x(i),x(j),x(v)$.
3. Table 1, with graph, in about half of the cases GCN outperforms all higher order methods. Any intuition on why? When should we expect higher topology to matter?
4. Is the "sim" function cosine similarity?

---

> ### Author Response · Authors · 2023-11-17
>
> ## Weaknesses
>
> 1. (Includes also the reply to Question 2.) Eq. (6) can be used for polygons of arbitrary size $k$. We initially presented it for the triangle case ($k=3$) just to provide a simple example. We made the choice to decompose the “$k$-th order” similarity as a sum of pairwise similarities to enhance the scalability of our method (trivially, a sum can be parallelized). However, it can be arbitrarily designed, as long as it is a similarity measure, i.e. it is higher if the involved edge features are similar. We rewrote Eq. (6) in its general form for a polygon of arbitrary size $k$ and included the above considerations in the revised paper.
>
> 2. The reviewer is right, indeed we conducted extensive ablation studies on the maximum inferrable polygon size $K_{max}$ (Tables 8 and 9  in the Appendix of the revised paper). We performed experiments using values for $K_{max} > 5$ but did not observe improvements. We also observed that very few times the inferred 1-skeletons presented cycles of lengths greater than 4 or 5. Although it has been shown that MP at the polygon level mitigates the GNNs’ limitation in handling long-range interactions (see e.g. Bodnar et al., Giusti et al.), we agree that future research on Latent Topology Inference could explicitly focus on this issue, given the promising results of this work.
>
>  3. It is true that, on the homophilic datasets, the results of the DCM with input graph are slightly worse than the other well-known baselines. We argue this is an expected behavior, because those models inherently rely on the homophily assumption. Our results are also consistent with previous literature on LGI (e.g. DGM). On the other hand, DCM consistently shows superior performance in the heterophilic settings when the graph is provided, whereas the performance of the other “homophily-based” methods tends to drop. When the input graph is not provided, the performance of the DCM is consistently superior across all the tested datasets. Finally, please note that DCM is a general framework allowing to employ other more complex GNNs and TopologicalNNs as backbones at the node and edge levels instead of the simpler GCN and CCCN we used in our paper for the sake of simplicity. This feature represents an exciting future direction.
>
> ## Questions
>
> 1. In Table 3 of Appendix B of the revised paper (reported also in the reply to Reviewer ims6), we show a breakdown of the inference execution times on two reference datasets, Cora and Texas, for GCN, GCN+CCCN, DGM, and DCM,  based on the (macro-)operations they require. As we can see from this empirical analysis and as we could expect from the complexity analysis we report in Appendix B, the lifting operation (computing the cycles and lifting the node embeddings) and cell sampling (computing the similarities among edge embeddings and applying the $\alpha$-entmax) are the main computational bottlenecks. The lifting operation needs to be performed in every complex-based architecture (we show DCM and GCN+CCCN, but for CWN would be the same); architectures that do not perform LTI (CWN or GCN+CCCN) can mitigate this problem (only) on transductive tasks by computing the cycles offline. Additional solutions w.r.t. the ones presented in point (b) of Appendix B could be accumulating the gradients and inferring the graph/cell-complexes once every $t$ iteration rather than after every optimization step. The cell sampling bottleneck, unlike the lifting, is not given by any technical requirement, but it is just related to our actual implementation. To keep the code readable and reproducible for (mainly) academic purposes, we use basic data structures and functions, e.g. we store the cells in lists that we parse. However, the bottleneck could be mitigated by optimizing the code. We are currently working on this and we will soon update our repo, hopefully before the end of the rebuttal period.
>
> 2. Please see Weakness 1.
>
> 3. About the "why", as explained in detail above,  we expect the results of DCM to be comparable to the other baselines on the homophilic datasets. About the “when”, several works tried to give a technical answer. To name a few,  some works showed that working on these spaces enhances the expressivity and the handling of long-range interactions (Bodnar et al., Giusti et al.), while some other works leveraged a Signal Processing perspective (Barbarossa et al., Yang et al., Calmon et al.) to prove that working on these spaces provides more versatile and sophisticated frequency-based tools. In general, as it often happens with DL, there is no unique answer. As long as multiway (group, high order, beyond pairwise) interactions play a crucial role in the adopted data, then higher topology will significantly help.
>
> 4. The “sim” function is any (pseudo-)similarity measure. Cosine similarity could be employed as well. In our experiments, we use minus the square distance among the embeddings. We clarified this in the revised paper.

---

> > ### Comment · Reviewer_XizW · 2023-11-21
> >
> > Thank you for the revisions and explanation. The run times in Table 3 are very good. The paragraph on complexity is also helpful. And thanks for providing the code. I have no further questions at this point.

---

### Official Review · Reviewer_ims6 · 2023-11-04

**Soundness:** 2 fair
**Presentation:** 3 good
**Contribution:** 3 good
**Rating:** 8
**Confidence:** 2

**Summary:**

This paper deals with the learning of high-order cell complexes with sparse and not regular graph topology.

The key is to introduce a learnable function that computes cell probabilities in the complex and integrate with cell complex message-passing network layers in a scalable way. The model achieved improve test accuracy in both homophilic and heterophilic graph node classification benchmarks. However, the improvements seem incremental.

**Strengths:**

The key is to introduce a learnable function that computes cell probabilities in the complex and integrate with cell complex message-passing network layers in a scalable way. The model achieved improve test accuracy in both homophilic and heterophilic graph node classification benchmarks.

**Weaknesses:**

In empirical studies, the improvements on accuracy seem incremental.

**Questions:**

What is the time efficiency of DCM, comparing with the baselines?

---

> ### Author Response · Authors · 2023-11-17
>
> ## Weaknesses
>
>  1. It is true that, on the homophilic datasets, the results of the DCM with provided graph are slightly worse than the presented well-known graph-based and complex-based architectures. We argue this is an expected behavior, because those architectures inherently rely on the homophily assumption, i.e. they employ diffusion/message-passing schemes that work better when connected nodes have similar labels. Our results are also consistent with previous literature on latent graph inference (e.g. DGM). On the other hand, DCM consistently shows superior performance in the heterophilic settings when the graph is provided, whereas the performance of the other “homophily-based” methods tends to drop. When the input graph is not provided, the performance of the DCM is consistently superior across all the tested datasets. To conclude, we want to stress that one should not consider DCM only as a novel effective GNN architecture. The main novelty of DCM is in it being (to our knowledge) the first model capable of Latent Topology Inference, i.e. learning of higher-order latent interactions modeled as combinatorial topological spaces (regular cell complexes, in our case). DCM is able to overcome many of the Latent Graph Inference's (and Topological Deep Learning's) limitations and achieve SOTA or near-SOTA results across multiple datasets. In addition, please note that DCM is a general framework allowing to employ other more complex GraphNN and TopologicalNN architectures as backbones at the node and edge levels instead of the simpler GCN and CCCN we used in our paper for the sake of simplicity (Remark 5). This feature represents an exciting direction for future work.
>
> ## Questions
>
> 1. In Table 3 of Appendix B of the revised paper, reported below, we show a breakdown of the inference execution times (in seconds) on two reference datasets, Cora and Texas, for GCN, GCN+CCCN, DGM, and DCM,  based on the (macro-)operations they require. As we can see from this empirical analysis and as we could expect from the complexity analysis we report in Appendix B, the lifting operation (computing the cycles and lifting the node embeddings) and cell sampling (computing the similarities among edge embeddings and applying the $\alpha$-entmax) are the main computational bottlenecks. The lifting operation needs to be performed in every complex-based architecture (we show DCM and GCN+CCCN, but for CWN would be the same); architectures that do not perform LTI (CWN or GCN+CCCN) can mitigate this problem (only) on transductive tasks by computing the cycles offline. Additional solutions w.r.t. the ones presented in point (b) of Appendix B could be accumulating the gradients and inferring the graph/cell-complexes once every $t$ iteration rather than after every optimization step. The cell sampling bottleneck, unlike the lifting, is not given by any technical requirement, but it is just related to our actual implementation. To keep the code readable and reproducible for (mainly) academic purposes, we use basic data structures and functions, e.g. we store the cells in lists that we parse. However, the bottleneck could be mitigated by optimizing the code. We are currently working on this and we will soon update our repo, hopefully before the end of the rebuttal period.
>
> |                       |          | Graph MP | Graph Sampling | Lifting | Cell Sampling | Cell MP |
> |-----------------------|----------|----------|----------------|---------|---------------|---------|
> | **Cora**              | GCN      | 5e^-5    | -              | -       | -             | -       |
> |                       | GCN+CCCN | -        | -              | 2e^-1   | -             | 2e^-4   |
> |                       | DGM      | 6e^-4    | 4e^-3          | -       | -             | -       |
> |                       | DCM      | 1e^-5    | 5e^-3          | 1e^-1   | 2e^-2         | 5e^-4   |
> | **Texas**             | GCN      | 2e^-5    | -              | -       | -             | -       |
> |                       | GCN+CCCN | -        | -              | 1e^-2   | -             | 2e^-4   |
> |                       | DGM      | 3e^-5    | 9e^-4          | -       | -             | -       |
> |                       | DCM      | 1e^-5    | 9e^-4          | 3e^-2   | 7e^-2         | 4e^-5   |

---

### Author Response · Authors · 2023-11-17

We would like to thank all reviewers for their evaluation of the paper. We are glad they agree on the novelty of the idea, the extensive evaluation we provide, and our analysis of the limitations (Reviewers XizW, EucF).  A point-by-point reply to all the comments is individually given in the sequel. We reply to comments in the same order they were presented by the reviewers, and we also group them by "Weaknesses" and "Questions". All the changes in the revised manuscript appear in blue color, to facilitate checking. We summarize here the main points.

**Major modifications of the paper**: based on the reviewers’ comments, we have clarified several points in Section 2 and Section 3 that were partially unclear (e.g., on our choice of similarity measure); we have also included an analysis of execution times in Appendix B and a complete reproducibility statement.

**Performance of DCM when a graph is provided**:  the reviewers raised the concern that DCM does not outperform the baselines when a graph is provided, especially if the underlying graph has a strong level of homophily. We would like to stress this is not a weakness, but rather an *expected* result, which is also in line with existing literature on graph inference and graph rewiring. In these situations, a standard message-passing network, e.g., GAT, is already very close to the optimum, as the input graph is perfectly aligned with its computational graph. Instead, our method performs significantly better in situations where traditional message-passing GNNs suffer e.g. due to heterophily, and consistently outperforms the baselines. We also note that DCM can be easily combined with other message-passing or existing graph-based and complex-based models, which may potentially improve its performance even further.

## References

Bodnar, Cristian, et al. "Weisfeiler and Lehman go cellular: Cw networks." Advances in Neural Information Processing Systems 34 (2021): 2625-2640.

Barbarossa, Sergio, and Stefania Sardellitti. "Topological signal processing over simplicial complexes." IEEE Transactions on Signal Processing 68 (2020): 2992-3007.

Yang, Maosheng, and Elvin Isufi. "Convolutional Learning on Simplicial Complexes." arXiv preprint arXiv:2301.11163 (2023).

Yang, Maosheng, Elvin Isufi, and Geert Leus. "Simplicial convolutional neural networks." ICASSP 2022-2022 IEEE International Conference on Acoustics, Speech and Signal Processing (ICASSP). IEEE, 2022.

Calmon, Lucille, Michael T. Schaub, and Ginestra Bianconi. "Higher-order signal processing with the Dirac operator." 2022 56th Asilomar Conference on Signals, Systems, and Computers. IEEE, 2022.

Giusti, Lorenzo, et al. "CIN++: Enhancing Topological Message Passing." arXiv preprint arXiv:2306.03561 (2023).

---

### Author Response · Authors · 2023-11-21

Dear Reviewers and Area Chair,
The deadline for the discussion phase is close.

We want to thank the reviewers for their comments, which undoubtedly helped to improve our work. We also tried our best to address the raised concerns.

We are still waiting for replies from reviewers ims6 and XizW. Could you please parse our responses and let us know if there is anything that we can do to further improve the paper?

Thanks a lot,
The authors

---

### Meta-Review · Area_Chair_gpLr · 2023-12-08

**Metareview:**

In this submission, the authors proposed a new differentiable cell complex module for embedding structured data with high-order topological information. The proposed model leads to a new message-passing mechanism, capturing multi-way interactions between data points. Experiments on node classification tasks demonstrate the potential of the proposed model to some extent.

Strengths: (a) The motivation of the model is clear and reasonable, and the paper is well-written. (b) All three reviewers appreciate the authors' efforts in the rebuttal phase. (c) The model architecture is novel and reasonable.

Weaknesses: In my opinion, the experimental results on node classification are not very convincing because the proposed method did not consistently outperform the baselines. Although it is not a severe problem considering the technical contribution of this submission, I hope the authors can add more experiments in the final version, e.g., testing the proposed method in more representative datasets (e.g., heterogeneous graphs, multi-graphs, or any other structured data with high-order interactions).

**Justification For Why Not Higher Score:**

The proposed method is novel, and the technical contribution of this submission is sufficient.
However, the experimental part should be enhanced further.

**Justification For Why Not Lower Score:**

All three reviewers scored this submission 8 points consistently, leading to clear acceptance.

---

### Decision · Program_Chairs · 2024-01-16

Accept (poster)